# Simulation Model of Reactive Nitrogen Species in an Urban Atmosphere using a Deep Neural Network: RNDv1.0

Junsu Gil[1], Meehye Lee[1*], Jeonghwan Kim[2], Gangwoong Lee[2], Joonyoung Ahn[3], Cheol-hee Kim[4]

[1] *Department of Earth and Environmental Sciences, Korea University, Seoul, South Korea*

[2] *Department of Environmental Science, Hankuk University of Foreign Studies, Yongin, South Korea*

[3] *Air Quality Forecasting Center, Climate and Air Quality Research Department, National Institute of Environmental Research (NIER), Incheon, South Korea*

[4] *Department of Atmospheric Science, Pusan National University, Busan, South Korea*

\* *Corresponding author*: Meehye Lee (meehye@korea.ac.kr)

**Abstract**

Nitrous acid (HONO), one of the reactive nitrogen oxides ($NO_y$), plays an important role in the formation of ozone ($O_3$) and fine aerosols ($PM_{2.5}$) in the urban atmosphere. In this study, a new simulation approach to calculate HONO mixing ratios using a deep neural technique based on measured variables was developed. The 'Reactive Nitrogen species simulation using Deep neural network (RND)' has been implemented in Python. The first version of RND (RNDv1.0) was trained, validated, and tested with HONO measurement data obtained in Seoul during the warm months from 2016 to 2019.

A k-fold cross validation and test results confirmed the performance of RND v1.0 with an Index Of Agreement (IOA) of 0.79 ~ 0.89 and a Mean Absolute Error (MAE) of 0.21 ~ 0.31 ppbv. The RNDv1.0 adequately represents the main characteristics of the measured HONO and thus, RND v1.0 is proposed as a supplementary model for calculating the HONO mixing ratio in a polluted urban environment.

## 1. Introduction

Surface ozone ($O_3$) pollution has been reported to be worsen over continental areas (Arnell et al., 2019;Monks et al., 2015;Varotsos et al., 2013;IPCC, 2014). In particular, a warmer climate is expected to increase surface $O_3$ concentrations and peak levels in polluted regions, depending on its precursor levels (IPCC 2021). As one of the short-lived climate pollutants (SLCPs), $O_3$ also interacts with the global temperature via positive feedback (Shindell et al., 2013;Myhre et al., 2017;Stevenson et al., 2013). Therefore, it is imperative to accurately predict the mixing ratios and variations of surface $O_3$. While operational models such as community multiscale air quality (CMAQ) have been used widely for this purpose, uncertainties still arise from poorly understood chemical mechanisms involving reactive nitrogen oxides (NOy) and volatile organic compounds (VOCs), and lack of their measurements (Mallet and Sportisse, 2006;Canty et al., 2015;Akimoto et al., 2019;Shareef et al., 2019;Cheng et al., 2022).

In the urban atmosphere, $NO_y$ typically includes $NO_x$ ($NO + NO_2$), HONO, $HNO_3$, organic nitrates (e.g., PAN), $NO_3$, $N_2O_3$, and particulate $NO_3^-$. These species are produced and recycled through photochemical reactions until they are removed through wet or dry deposition (Liebmann et al., 2018;Brown et al., 2017;Wang et al., 2020;Li et al., 2020). $NO_y$ play an important role in critical environmental issues concerning the Earth's atmosphere, spanning from local air pollution to global climate change (Sun et al., 2011;Ge et al., 2019). The oxidation of NO to $NO_2$, and finally to $HNO_3$, is the backbone of the chemical mechanism producing ozone ($O_3$) and $PM_{2.5}$ (particulate matter of size $\leqslant$ 2.5 μm), and it determines the oxidization capacity of the atmosphere. Recently, as $O_3$ has increased along with a decrease in $NO_x$ emission over many regions including East Asia, interest in the heterogeneous reaction of reactive nitrogen oxides, which is yet to be understood, has been newly raised (Brown et al., 2017;Stadtler et al., 2018). Currently, the lack of measurement of individual $NO_y$ species hindered a comprehensive understanding of the heterogeneous reactions (Anderson et al., 2014;Wang et al., 2017b;Chen et al., 2018b;Akimoto and Tanimoto, 2021;Stadtler et al., 2018).

In particular, there are growing number of evidence for heterogeneous formation of HONO in relation to high $PM_{2.5}$ and $O_3$ occurrence in urban areas (e.g., (Li et al., 2021b). As

an OH reservoir, HONO will expedite the photochemical reactions involving VOCs and NOx
in the early morning, leading to $O_3$ and fine aerosol formation. Nonetheless, its formation
mechanism has not been elucidated clearly enough to be constrained in conventional
photochemical models. In addition to the reaction of NO with OH (Bloss et al., 2021), various
pathways of HONO formation have been suggested from laboratory experiments, field
measurements, and model simulations: direct emissions from vehicles (e.g., (Li et al., 2021a)
and soil (e.g.,(Bao et al., 2022), photolysis of particulate nitrate (e.g., (Gen et al., 2022),
heterogeneous conversion of $NO_2$ on various aerosol surfaces (e.g., (Jia et al., 2020), ground
surface (e.g.,(Meng et al., 2022), and microlayers of sea surface (e.g., (Gu et al., 2022). Among
these, the heterogeneous reaction mechanism on the surface is of major interest in the recent
HONO study.
HONO has been measured mostly during intensive campaigns in urban areas using
various techniques such as a long path absorption photometer (LOPAP) (Kleffmann et al.,
2006;Xue et al., 2019), chemical ionization mass spectrometry (CIMS) (Levy et al.,
2014;Roberts et al., 2010), ion chromatography (IC) (VandenBoer et al., 2014;Gil et al.,
2020;Ye et al., 2016;Xu et al., 2019), and quantum cascade tunable infrared laser differential
absorption spectrometry (QC-TILDAS) (Lee et al., 2011;Gil et al., 2021). Of these methods,
QC-TILDAS has served as a reference for intercomparison of measurement data from different
techniques due to high time resolution and stability (Pinto et al., 2014). These studies reported
the maximum HONO of several ppb levels at nighttime. In comparison, the model captured at
most 67~90 % of the observed HONO in megacities such as Beijing (Tie et al., 2013;Liu et al.,

82    2019).

In recent years, Machine Learning (ML) method has been adopted in the atmospheric
science for pattern classification (e.g. New Particle Formation event) and forecasting and
spatiotemporal modelling of $O_3$ and $PM_{2.5}$ (Arcomano et al., 2021;Shahriar et al.,
2020;Krishnamurthy et al., 2021;Cui and Wang, 2021;Joutsensaari et al., 2018;Chen et al.,
2018a;Kang et al., 2021). Among ML methods, the Neural Network (NN) architecture is widely
used owing to its powerful ability to process large amounts of data, allowing improvement in
the performance of conventional models through being integrated with physical equations
(Reichstein et al., 2019;Schultz et al., 2021). As a NN architecture, a multi-layer artificial neural
network, referred to as a Deep Neural Network (DNN), employs a statistical method that learn
non-linear relations in data and obtain the optimum solution for the target species without prior
information on the physicochemical processes. DNN has advantages over other NN architecture
such as Convolution NN (CNN) or Long-Short Term Memory (LSTM) because it works well
for discrete spatiotemporal data. In general, the performance of DNN is similar to or better than
other ML methods for small number of data as well as large data set (Baek and Jung, 2021;Dang
et al., 2021;Sumathi and Pugalendhi, 2021).

98        When the DNN method is applied to atmospheric chemical constituents, it requires

large amount of data for training and thus, the size of measurement data becomes a limiting
factor for trace species such as HONO, which are not routinely measured such as $O_3$ or $PM_{2.5}$.
In this regard, the daily average HONO mixing ratio was attempted to be estimated using
ensemble ML models with satellite measurements (Cui and Wang, 2021). In comparison, the
hourly HONO mixing ratio was calculated using a simple NN architecture with measured
variables, which were thought to be deeply involved in the formation of HONO (Gil et al.,
2021). The accuracy of the hourly HONO estimated from input variables such as aerosol surface
areas and mixed layer height was better than the daily HONO estimate.
The aim of this study is to develop a user-friendly 'Reactive Nitrogen species simulation using
DNN' model (RNDv1.0) that estimates HONO mixing ratios from real-time measurements of
criteria pollutants and meteorological parameters and is ultimately to be incorporated into
operational models that forecast urban air quality. Since this study is the first attempt to
calculate the HONO mixing ratio using RNDv1.0, the entire construction process is described
in detail, and the performance is evaluated by comparing the results with simulations using a
commonly used model and observations over several years.

**2. Model description**

The development of RNDv1.0 model follows the systematic steps similar to a general
machine learning model construction workflow, including collecting data, preprocessing data,
building the DNN, training and validating the model, and testing the performance of the model
(Figure 1). The RNDv1.0 was written in Python and necessary libraries to build and operate
RNDv1.0 are listed in Table 1. The dataset used to train-test-validation can be downloaded from
Gil et al., 2021.

**2.1. Collection of measurement data for model construction**

126         As the first step constructing the RNDv1.0, measurement data were obtained including
HONO, reactive gases, and meteorological parameters. It is noteworthy that the HONO
measurement data is for model construction and is not required to run the RND model. The
HONO mixing ratio was measured using a Quantum Cascade – Tunable Infrared Laser
Differential Absorption Spectrometer (QC-TILDAS) system in Seoul during May–June 2016,
June 2018, and April-June 2019 (Lee et al., 2011;Gil et al., 2021). When testing and evaluating
atmospheric HONO measurement methods, QC-TILDAS has been chosen as the reference
method for comparing ambient HONO mixing ratios measured using several different
techniques owing to its advantages of low detection limits (~ 0.1 ppbv) and high temporal
resolution (Pinto et al., 2014). More details on measurements can be found elsewhere (Gil et
al., 2021).

137         HONO was measured at Olympic Park (37.52°N, 127.12°E) during the Korea-United
States Air Quality (KORUS-AQ) study in 2016 (Kim et al., 2020;Gil et al., 2021), at the campus
of Korea University (37.59°N, 127.03°E) in 2018, and at the site near the campus (37.59°N,
127.08°E) in 2019 (NIER, 2020) (Figure S1). Although measurements were made at three sites,
$O_3$ and $PM_{2.5}$ levels have been known to be greatly influenced by the synoptic circulation
throughout the Korean peninsula (Peterson et al., 2019;Jordan et al., 2020), and the Korea
University campus and Olympic Park have served as measurement sites representing the air
quality of Seoul. In addition to HONO, trace gases including $O_3$, $NO_2$, CO, and $SO_2$ and
meteorological parameters including temperature (T), relative humidity (RH), wind speed (WS)
and direction (WD) were measured. Note that HONO was not significantly correlated with any
of these variables (Figure S2). The measurement statistics are presented in Table 2 and Table
S1. Briefly summarizing, the 10[th] and 90[th] percentile mixing ratios of HONO, $NO_2$, and $O_3$ are
0.3 ppbv and 1.9 ppbv, 10.7 ppbv and 48.2 ppbv, and 12.0 ppbv and 80.9 ppbv, respectively for
the entire experiment periods.

## 2.2. Data preprocessing


In the next step, the observation data set was prepared for RNDv1.0 model construction.
As input variables, hourly measurements of chemical and meteorological parameters are used,
including the mixing ratios of $O_3$, $NO_2$, CO, and $SO_2$, along with temperature (T), relative
humidity (RH), wind speed (WS), wind direction (WD), and solar zenith angle (SZA) to
estimate the target species, HONO, as the output. Wind direction in degrees were converted to
a cosine value for continuity. As a last step in data processing, missing values were filtered out
from the input dataset. Finally, 50.7 % of all available measurement data (1636) were used to
construct the RNDv1.0 in this study.
Since the measurements of these nine variables vary over a wide range in different units,
they were normalized to avoid bias during the calculations. Among the widely used
normalization methods, '*min-max scaling*' method was adopted and input variables were
normalized against the minimum and maximum values in this study (Eq. 1):

$$x_{sca} = \frac{x_{raw} - F_2(X)}{F_1(X)},$$      (1)

where $x_{raw}$ is raw data, $x_{sca}$ is scaled value, and $F_1$ and $F_2$ are scale factors of input
variable (X), which are listed in Table 2.

## 2.3. Neural network architecture and hyperparameters


At this stage, the network is built to calculate HONO using those input variables. The
RNDv1.0 is composed of five hidden layers (Figure 2), which employed an exponential linear
unit (ELU) as an activation function (Eq. 2).

$$\text{ELU}: \phi(x) = \begin{cases} e^x - 1 \ (x < 0) \\ \quad x \ (x \geq 0) \end{cases}.$$     (2)

In a DNN, an activation function creates a nonlinear relationship between an input
variable and an output variable. When constructing a DNN model, an ELU has the advantage
of a fast-training process and better performance in handling negative values than other
activation functions (Wang et al., 2017a;Ding et al., 2018). In addition, the mean squared error
and Adam optimizer were applied as loss function and optimize function, respectively. The
learning rate, epoch, and batch were set to 0.01, 100, and 32, respectively.

**2.4. Model training**

The RNDv1.0 model was trained, validated, and tested with HONO measurements obtained
during May ~ June in 2016 and 2019, in June 2018, and in April 2019, respectively (Figure 3).
The number of data used for train, validation, and test were 1122, 381, and 133, respectively.
With the hyperparameters specified in previous section, the performance of the model was
firstly validated using the k-fold cross-validation method, which is especially useful when the
size of dataset is small (Bengio and Grandvalet, 2003). In the k-fold cross-validation method
(Figure 3), the entire data is randomly divided into k subsets, of which k-1 sets were used for
training and the rest one was used for validation. k was set to 5 in this study. The accuracy was
determined by Index Of Agreement (IOA) and Mean Absolute Error (MAE) expressed by the
following equation (Eq. 3, Eq. 4):

$$\text{IOA} = 1 - \frac{\sum_{i=1}^{n}(O_i - P_i)^2}{\sum_{i=1}^{n}(|P_i - \bar{O}| + |O_i - \bar{O}|)^2},$$     (3)
$$\text{MAE} = \frac{\sum_{i=1}^{n}|O_i - P_i|}{n},$$     (4)

where $O_i$, $P_i$, $\bar{O}$, and n are the observed value, predicted value, average of the observed
values, and number of nodes, respectively. The overall accuracy of

As IOA and MAE vary according to the number of nodes, they were calculated for the measured (HONO$_{obs}$) and calculated (HONO$_{mod}$) mixing ratios by varying the number of nodes from 0 to 100 in each hidden layer. The best performance was found with 41 nodes, with which the averaged IOA and MAE were $0.89 \pm 0.01$ (mean $\pm$ standard deviation) and $0.31 \pm 0.02$ ppbv, respectively (Figure 4). The high level of IOA and low MAE demonstrates that the performance of RNDv1.0 model is adequate, and it is capable of simulating the ambient HONO mixing ratio using the routinely measured criteria pollutants and meteorological parameters. In particular, MAE was commensurate with the detection limit of HONO measurement.

After the network validation, HONO mixing ratio was calculated for May ~ June in 2016 and 2019, and the model results were compared with the measured values (Figure 5). The average mixing ratios of measured and calculated HONO was 0.94 ppbv and 0.89 ppbv in 2016, and 1.02 ppbv and 0.96 ppbv in 2019, respectively. The MAE and IOA of the measurement and calculation were 0.27 ppbv and 0.90 in 2016, and 0.29 ppbv and 0.91 in 2019, respectively, demonstrating the ability of the RNDv1.0 to simulate ambient HONO levels. In both cases, however, the model slightly underestimated the highest and lowest HONO mixing ratios, which is mainly due to the limited number of data used for training, but also related to the intrinsic nature of DNN. The model calculation well captured the diurnal variation of ambient HONO with a slight underestimation (Figure 6). In addition, the correlation between HONO$_{mod}$ and HONO$_{obs}$ was better in 2019 (MAE = 0.06 ppbv) than in 2016 (MAE = 0.08 ppbv). Since the MAE of the two cases was far below the detection limit of HONO measurements (~ 0.1 ppbv), the RNDv1.0 is considered suitable for simulating HONO in urban areas.

Next, the HONO calculated in RNDv.1.0 was compared with observations and results from CMAQ (Community Multi-scale Air Quality Model, v5.3.1) simulations during the KORUS-AQ study (May~June 2016) (Figure 7). More information on CMAQ modeling can be found elsewhere (Appel et al., 2021). While the results of RNDv.1.0 reasonably traced the observed variations (IOA = 0.90), the CMAQ severely underestimated the measured HONO concentration (IOA = 0.44). These results demonstrate the performance and efficacy of RNDv1.0 in calculating the ambient HONO mixing ratio that are poorly reproduced in conventional operating models.

## 2.5. Influence of input variables on HONO concentration

236

A simple bootstrapping test was conducted to evaluate the relative importance of the input variable to HONO concentration. In this analysis, each variable was set to zero and MAE was calculated as an evaluation metrics (Kleinert et al., 2021). Of nine input variables, $NO_2$ was found to have the most significant influence on HONO concentration, followed by RH, temperature, and solar zenith angle (Table 4). The highest MAE of 0.59 ppbv can be considered as the maximum uncertainty of RNDv1.0 due to the input variable.

The result of bootstrap test is in good agreement with those of our previous study (Gil et al., 2021), where more variables such as aerosol surface area and mixing layer height were incorporated into the model, highlighting the crucial role of precursor gases and heterogeneous conversion in HONO formation. Therefore, these results demonstrate that the RND model constructed from routinely measured criteria pollutants and meteorological parameters sufficiently captured the HONO variability in the urban atmosphere.

249

**2.6. Model validation and test**

251

Finally, the RND model was validated and tested against the measurement data obtained in June 2018 and April 2019. The calculated HONO mixing ratios are compared with those measured in Figure 8, and their MAE and IOA are listed in Table 3. The two sets of model performance test showed that the model reasonably traced what was observed. As the validation result of RND, the MAE and IOA of the calculated and measured in June 2018 are comparable to those of 2016~2019 result. However, the MAE and IOA of the April 2019 measurements were relatively poor compared to the validation results. Especially, the MAE of the April 2019 is about twice as high as those of validation.

In these two test periods, HONO levels were lower than those observed on validation days (Figure 8), and the model tended to overestimate high HONO concentrations. It is possibly due to the variability of HONO that is not fully captured by RNDv1.0 using 9 input variables. As stated above, heterogeneous reactions intimately involved in HONO formation are not considered in RNDv1.0. More importantly, the annual variability of criteria pollutants such as

PM$_{2.5}$ has increased in recent years. Particularly in 2019, the monthly average PM$_{2.5}$ mass concentration was lower in April (21 μg m$^{-3}$) than in May (29 μg m$^{-3}$), unlike normal years. Given that the test result is within the uncertainty range of the model that is primarily determined by NO$_2$ (Table 4), RNDv1.0 will be applicable to urban environments under various conditions.

## 3. Operation and application of RNDv1.0

The RNDv1.0 package is provided as an operational model, .h5 files that can be opened in Python. To run the RNDv1.0, the measurement data for nine input variables are required and need to be properly prepared as described in Section 2.2. A sample of preprocessed input dataset is provided as a .csv file (Dataset_for_model.csv). Once the input data is ready, open the RNDv1.0 with input data files using the code provided in the example (Figure S3). Then, RND v1.0 calculates and presents the HONO results as scaled values ($x_{sca}$), which will be finally converted to HONO mixing ratio (ppbv) by the two scale factors in Table 2 (Eq. 5):

$$HONO \ (ppbv) = HONO_{sca} \times F_1(HONO) + F_2(HONO). \tag{5}$$

The HONO calculated by Eq. 5 can be applied to an urban photochemical cycle simulation. It is already known that the photolysis of HONO is a major source of OH radicals in the early morning when the OH level is low, and this OH affects daytime O$_3$ formation through photochemical reactions with VOCs and NO$_x$, which are primarily emitted during morning rush hour in urban areas. In addition, the OH produced from HONO promotes the photochemical oxidation of SO$_2$ and VOCs, leading to aerosol formation. However, the HONO formation mechanism is still poorly understood, hindering O$_3$ and fine aerosols as well as HONO from being correctly simulated in conventional photochemical models.

The 0-Dimension Atmospheric Modelling (F0AM) utilizing the MCM v3.3.1 chemical reaction mechanisms (Wolfe et al., 2016), can be used to simulate the diurnal variation of O$_3$ with the measurements of several reactive gases (NO, NO$_2$, CO, HCHO, VOCs, and HONO).

Detailed information about F0AM can be found in
(https://sites.google.com/site/wolfegm/models) and in previous works published elsewhere
(Wolfe et al., 2016; Gil et al., 2020). When the F0AM model is run without HONO, it is not
able to reproduce the concentration and diel cycle of the observed $O_3$ (Figure 9). In comparison,
the model simulates the $O_3$ well within 2 ppbv when adding HONO, which is the product of
RND v1.0. This is mainly due to the missing OH produced by HONO photolysis in the early
morning. Its production rate is estimated to be 0.57 pptv $s^{-1}$, contributing approximately 2.28
pptv to OH budget during 06:00 ~ 11:00 (LST) (Gil et al., 2021). Given that OH is mainly
produced from the photolysis of $O_3$ under high sun, the early morning source of OH will
expedite the photochemical cycle involving $NO_x$ and VOCs, promoting $O_3$ and secondary
aerosol formation. Since the presence of HONO in the photochemical model allows for accurate
estimation of OH radicals, the incorporation of RNDv1.0 into conventional models will
improve their overall performance.

**4.  Summary and implications**

In this study, we developed the RND model to calculate the mixing ratio of $NO_y$ in an urban
atmosphere using a DNN along with measurement data. The target species of RNDv1.0 is
HONO, and its mixing ratio is calculated using criteria pollutants including $O_3$, $NO_2$, CO, and
$SO_2$, and meteorological variables including T, RH, WS, and WD, along with the SZA. These
variables are routinely measured through monitoring networks. The RNDv1.0 was trained and
validated using the HONO measurements obtained in Seoul by adopting a k-fold cross
validation method and tested with other HONO datasets measured using the same instrument.
The validation and test results demonstrate that RNDv1.0 adequately captures the characteristic
variation of HONO and confirms the efficacy of RND v1.0.
RNDv1.0 was constructed using the measurement made in a high $NO_x$ environment during
early summer (May–June). It is noteworthy that in this period, the HONO mixing ratio was
raised above 3 ppbv with the highest $O_3$ levels under stagnant conditions. If RNDv1.0 is applied
to areas under significant influence of outflows, the model possibly overestimates or
underestimate the level of HONO without detailed information such as nanoparticles. In the

previous study, the formation of HONO was shown to be intimately related with surface areas of submicron particles (Gil et al., 2021). Nevertheless, the HONO concentration produced from RNDv1.0 with routine measurements provides the benefit of relatively inexpensive test for measurement quality control, location selection, and supports the data used for traditional chemistry model based on the current knowledge of the urban photochemical cycle. Therefore, it is reasonable to argue that RNDv1.0 can serve as a supplementary tool for conventional forecasting models. As attempts are currently being made to estimate ground HONO from satellite observations (Clarisse et al., 2011;Theys et al., 2020;Armante et al., 2021), RNDv1.0 will also be useful for validating satellite-derived HONO by supplementing measurement data.

## 5. Acknowledgements

This study was supported by the National Research Foundation of Republic of Korea (2020R1A2C3014592) and Korea Institute of Science and Technology (KIST2E31650-22-P019).

## 6. Code availability

The RND model codes (.h5 files) with preprocessed sample data can be downloaded from (Gil, 2021).

## 7. Author contributions

JG and ML designed the manuscript and developed the model code. JK, GL, and JA provided HONO measurements and CK provided CMAQ model data. All the authors contributed to the manuscript.

**8. Competing interests**

The authors declare that they have no conflict of interest.


**Figures and Tables**

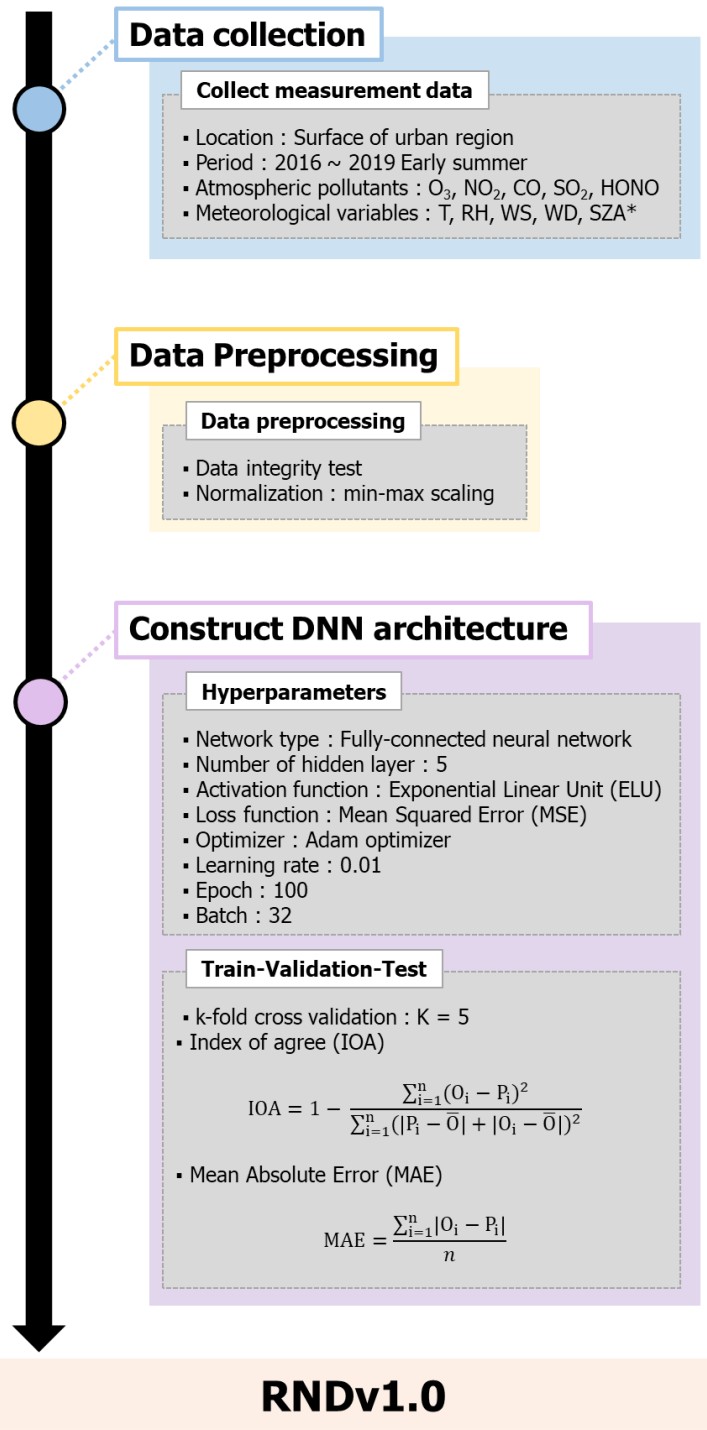

**Figure 1.** The main processes for configuring the RNDv1.0 (*: calculated values)


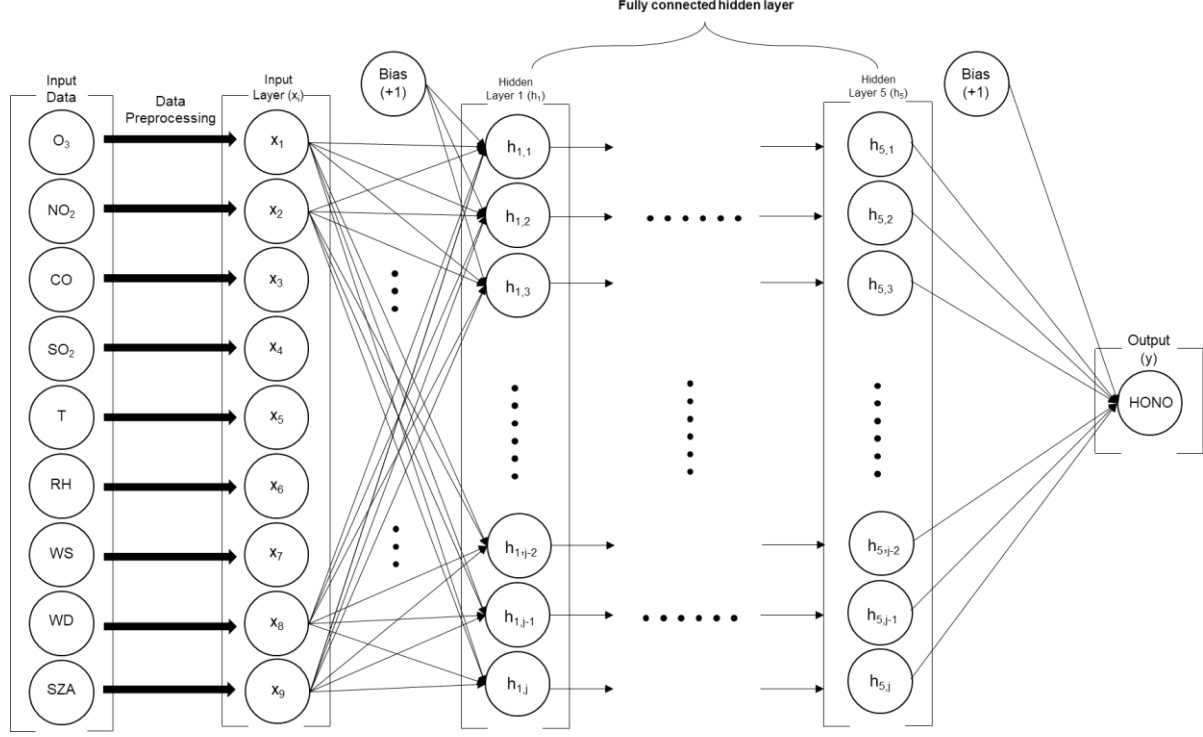

**Figure 2.** The structure of deep neural network built for RND v1.0.


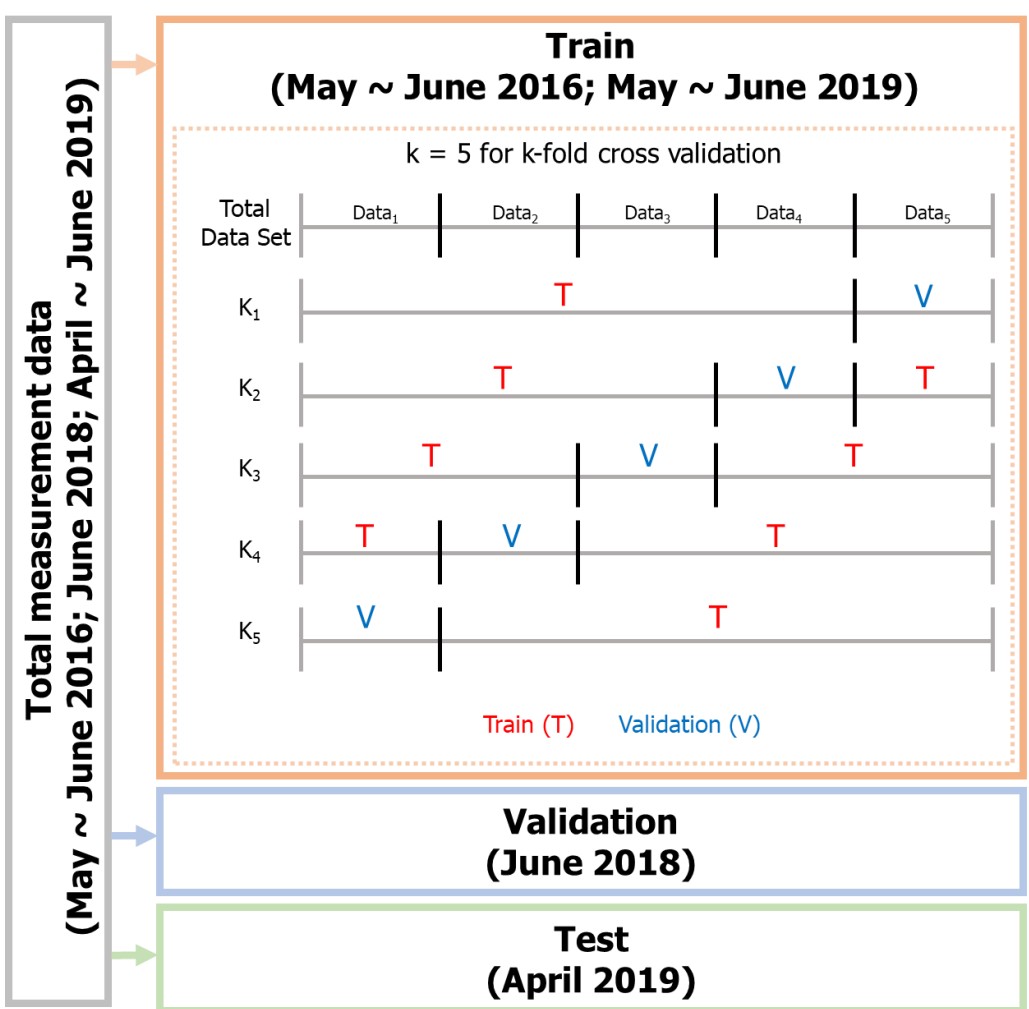


**Figure 3.** Design of training, validation, and test to build RNDv1.0 using measurement data. The k-fold cross validation were performed using randomly divided five subsets of training data set.



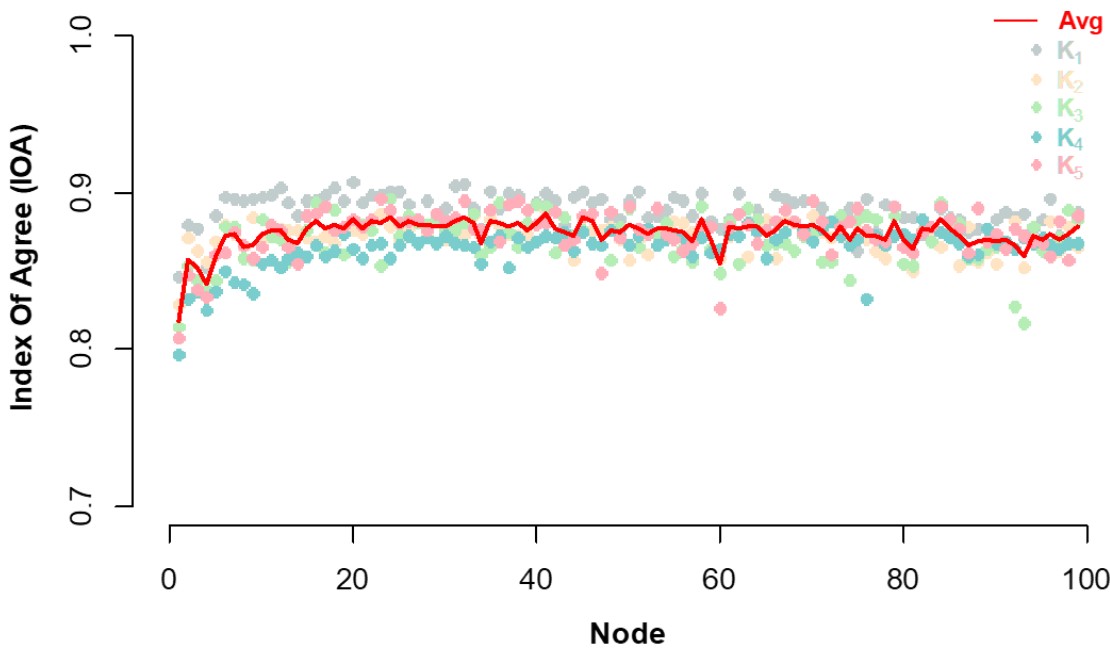


**Figure 4.** Index Of Agreement (IOA) for k-fold cross validation. Solid circle and red line
represent IOA for each validation (k=5) and the average of 5 validation sets at each node number.


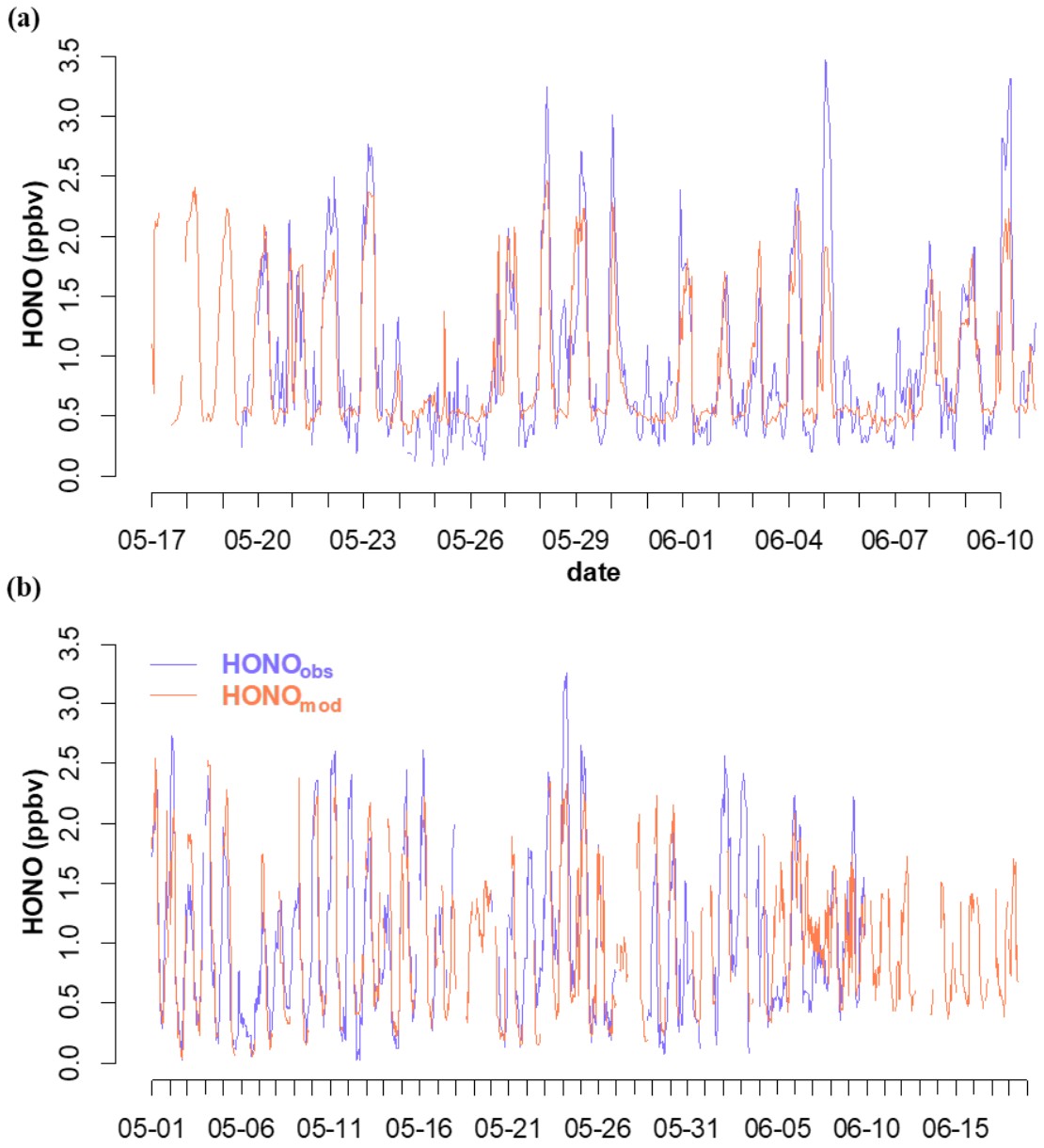


**Figure 5.** Comparison between the measured (HONO_obs) and calculated (HONO_mod) HONO
mixing ratios in Seoul during May~June in (a) 2016 and (b) 2019. The blue and red lines
indicate the measured and calculated HONO mixing ratio, respectively.

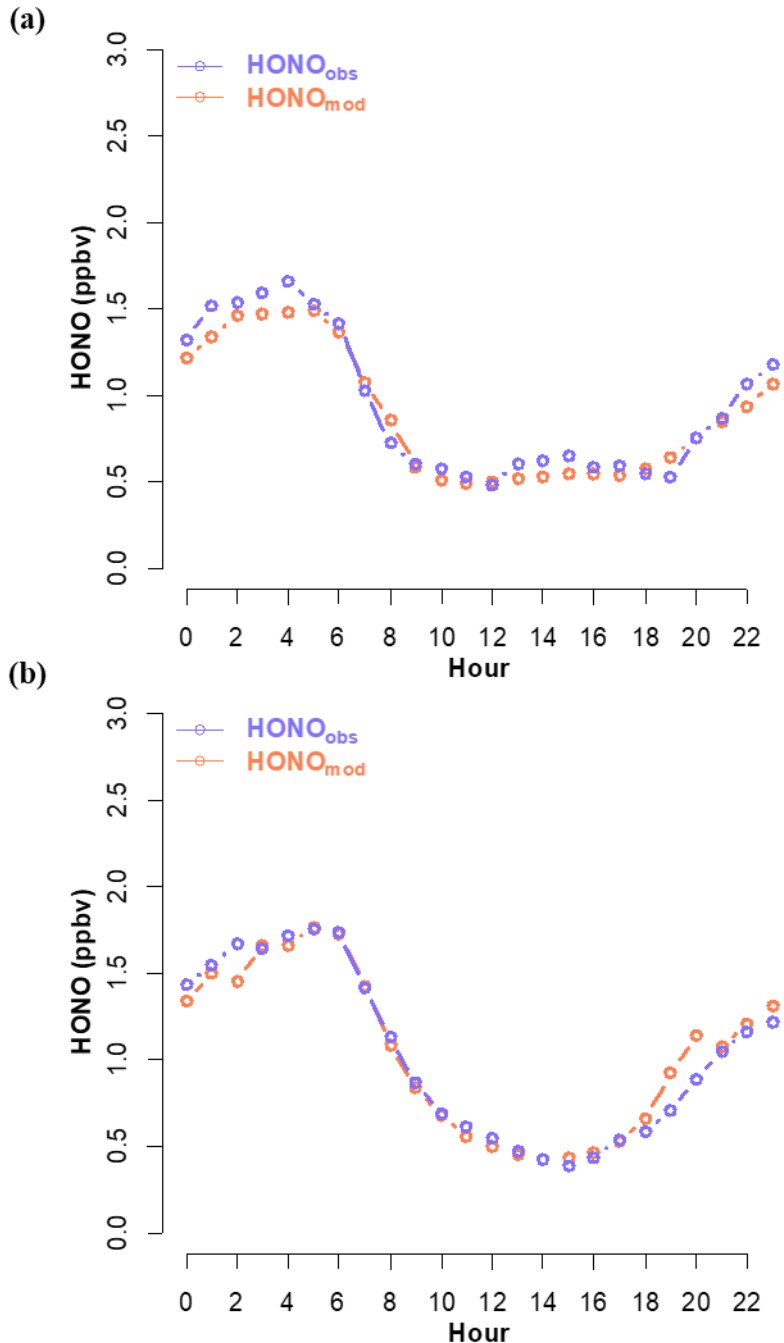


**Figure 6**. Average diurnal variations of the measured (HONO_obs) and the calculated (HONO_mod)
HONO mixing ratios in Seoul during May ~ June in (a) 2016 and (b) 2019.


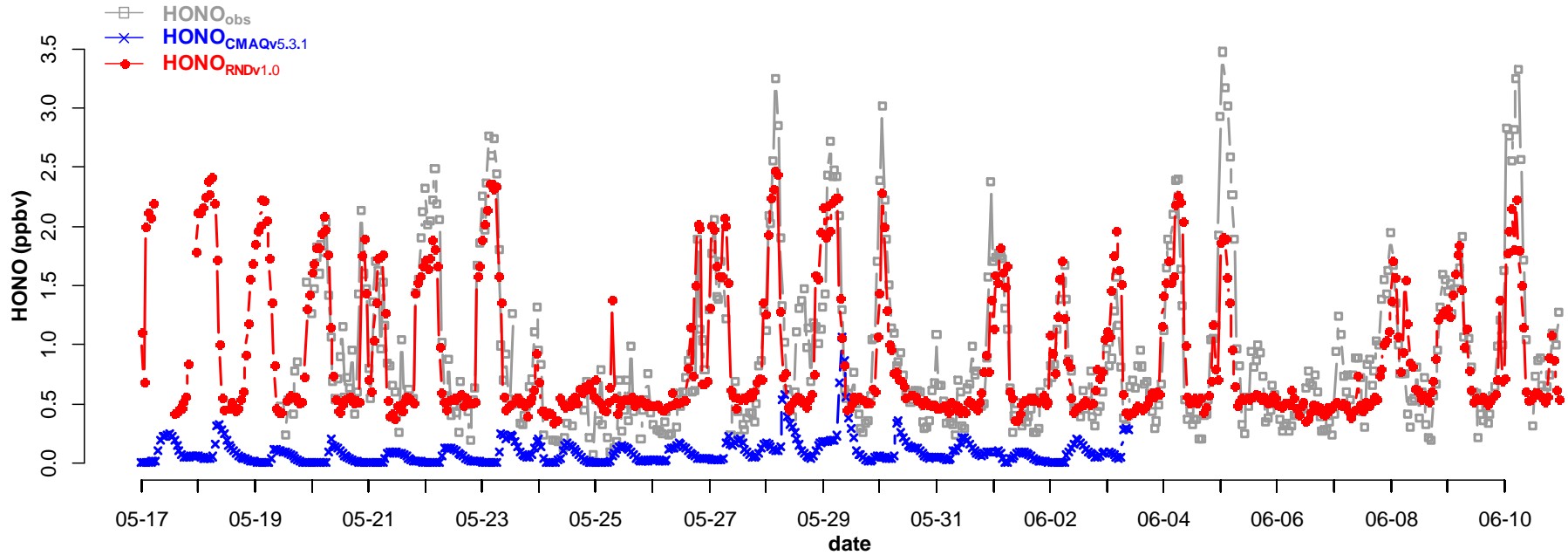


**Figure 7**. During the KORUS-AQ campaign (May-June 2016), HONO mixing ratios calculated using RNDv1.0 (red dot) are compared with those
observed (gray square) and calculated using CMAQv5.3.1 (blue cross).

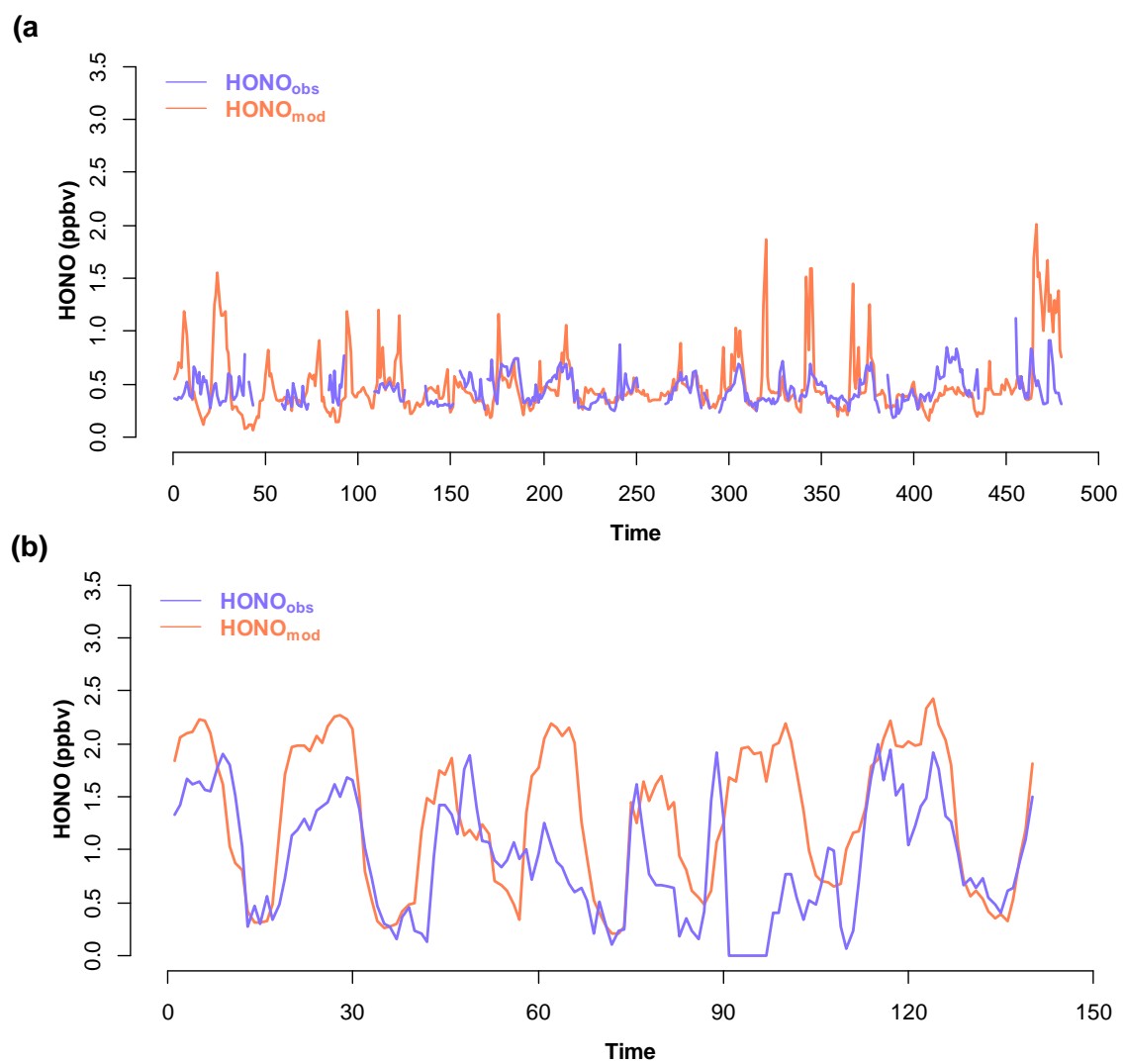

**(a)**

**(b)**


**Figure 8.** Comparison between the measured (HONO_obs) and calculated (HONO_mod) HONO
mixing ratios in Seoul during (a) June 2018 and (b) April 2019. The blue and red lines indicate
the measured and calculated HONO mixing ratio, respectively. The x axis indicates the hour
from the beginning of the experiment, which is (a) 00:00 on 1$^{st}$ June 2018 and (b) 00:00 on 12$^{th}$
April 2019.

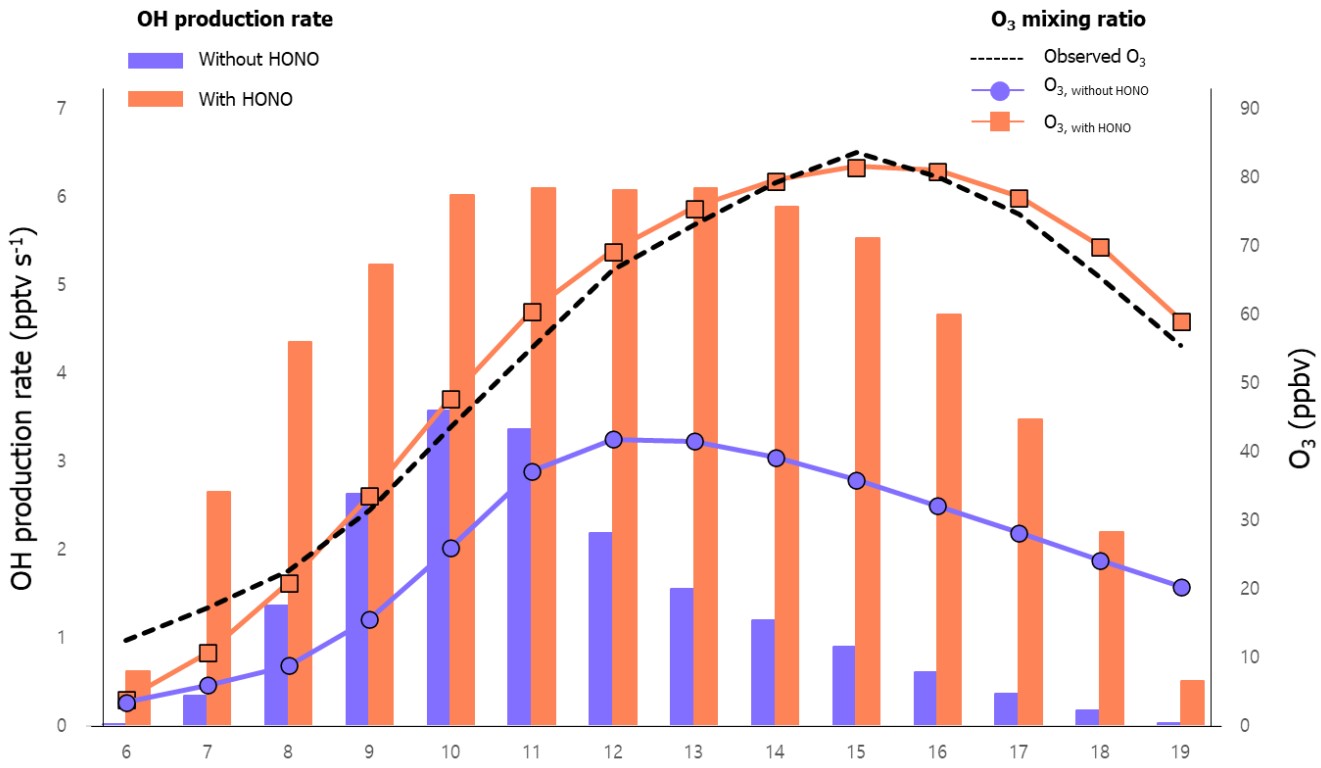


**Figure 9**. For June 2016, diurnal variations of $O_3$ (line) and OH production rate (bar) calculated

from the F0AM photochemical model with (orange) and without (blue) HONO estimated from

the RNDv1.0 model. The measured $O_3$ is compared with the calculated.


**Table 1.** Resources for constructing RND model.

|  | Version | Remark |
|---|---|---|
| Python | v3.8.3 | |
| CUDA | v10.1 | *If using GPU |
| CuDNN | v7.6.5 | *If using GPU |
| Tensorflow | v2.3.0 | *Python library* |
| Keras | v2.4.3 | *Python library* |
| Pandas | v1.0.5 | *Python library* |
| Numpy | v1.18.5 | *Python library* |

*GPU denotes graphic processing unit
**Table 2**. Input variables and their concentrations ($10^{th} \sim 90^{th}$ percentile), coverage, and scale
factors for RNDv1.0 model. Measurements were made in Seoul during May ~ June in 2016 and
402 2019.

| | $10^{th} \sim 90^{th}$ percentile (unit) | Coverage (%) | Scale Factor1 $(F_1)$* | Scale Factor 2 $(F_2)$** |
|---|---|---|---|---|
| **Input Variables** | | | | |
| $O_3$ | 12.1 ~ 90.4 (ppbv) | 95.5 | 204.738 | 0.842 |
| $NO_2$ | 11.0 ~ 48.6 (ppbv) | 80.6 | 79.925 | 2.375 |
| CO | 252 ~ 743 (ppbv) | 95.1 | 975.248 | 137.253 |
| $SO_2$ | 1.9 ~ 6.4 (ppbv) | 95.6 | 12.479 | 0.958 |
| Solar Zenith Angle | 22.7 ~ 118.4 (º) | 100.0 | 112.317 | 14.195 |
| Temperature | 15.9 ~ 26.7 (°C) | 99.4 | 24.240 | 8.610 |
| Relative Humidity | 29.2 ~ 79.1 (%) | 99.4 | 88.545 | 10.555 |
| Wind Speed | 0.2 ~ 3.7 (m/s) | 99.4 | 7.581 | 0.005 |
| Wind Direction | 45.4 ~ 287.5 (º) | 99.4 | 359.565 | 0.235 |
| **Output Variables** | | | | |
| HONO | 0.3 ~ 2.0 (ppbv) | 81.1% | 3.447 | 0.013 |

* Maximum – Minimum
** Minimum value

**Table 3.** The result of validation and test of RNDv1.0 model using measurement data.

| Measurement data | Validation | | Test | |
| --- | --- | --- | --- | --- |
| | MAE (ppbv) | IOA | MAE (ppbv) | IOA |
| May 2016* | 0.26 | 0.93 | | |
| June 2016* | 0.29 | 0.86 | | |
| June 2018 | 0.21 | 0.79 | | |
| April 2019 | | | 0.56 | 0.65 |
| May 2019* | 0.26 | 0.93 | | |
| June 2019* | 0.36 | 0.76 | | |

*Re-using the data already used for training

**Table 4.** The result of bootstrap test of measurement data used to train RNDv1.0 model. The
greater the MAE, the greater the influence of variable.

| Variable (X) | MAE (ppbv) |
|---|---|
| - | 0.28 |
| $O_3$ | 0.29 |
| $NO_2$ | 0.59 |
| CO | 0.37 |
| $SO_2$ | 0.34 |
| Solar zenith Angle (SZA) | 0.41 |
| Temperature (T) | 0.52 |
| Relative humidity (RH) | 0.52 |
| Wind speed (WS) | 0.34 |
| Wind direction (WD) | 0.29 |

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
