# Peer review of "Simulation Model of Reactive Nitrogen Species in an Urban Atmosphere using a Deep Neural Network: RNDv1.0"

_Geoscientific Model Development, 2021_

## Author Comment (AC1)

Review of gmd-2021-347

Simulation Model of Reactive Nitrogen Species in an Urban Atmosphere using a Deep Neural Network: RND v1.0

by Junsu Gil et al.

General comments:

This manuscript describes a new application of a simple feed forward neural network model to calculate HONO mixing ratios based on a set of other measured variables. While this is an interesting and worthwhile application, the paper lacks the necessary details in the description of the deep learning model and contains no ablation studies which are needed to provide the credibility in the results. I also question the validity of the cross validation and test cases that are discussed, because I doubt that these test cases are truly independent data samples. There is no proof of the generalisation capability of the model, so it may well be that this model fails completely if it were applied to measurement data obtained under different conditions.

In summary, this manuscript falls between "major revisions" and "reject". In computer science conferences it would be ranked "weak reject", which means the paper could be saved if the authors invest substantial work in rerunning their model several times and improving the text.

- We are grateful for your constructive and considerate comments. The point-by-point responses are given below, along with relevant parts of the revised manuscript, where all changes are marked in blue.

Specific comments:

Abstract: Confusing sentence after "In this study,". After reading 3 times I understood that you are resolving the acronym RND here, but this is well hidden. Suggestion: "In this study, a new simulation approach to calculate HONO mixing ratios using a deep learning technique based on measured variables was developed. The 'Reactive Nitrogen species Deep neural network' (RND) has been implemented in Python. It was trained, ..."

- As suggested, the abstract is rewritten for clarification.

- Line 17-22: In this study, a new simulation approach to calculate HONO mixing ratios using a deep neural technique based on measured variables wad developed. The 'Reactive Nitrogen species simulation using Deep neural network' (RND) has been implemented in Python. It was trained, validated, and tested with HONO measurement data obtained in Seoul during the warm months from 2016 to 2019.

Abstract: Why should RND be called a *supplementary* model? What does it supplement?

- As mentioned in your comment L.250/251, HONO mixing ratios estimated from the RNDv1.0 model can be used for various purposes. Finally, you agreed that RND was a 'supplementary' tool.

l.35: too vague "observational constraints on individual species". Does this refer to NOy compounds or any species involved in the tropospheric ozone production cycle?

- It refers to NOy species. This sentence is removed in the revised manuscript.

l.40: NOy has been the focus of attention already in the 1990s. See for example papers by Sandy Sillman et al. You may say "renewed attention".

- Yes, you are right. This sentence is meant to emphasize the heterogeneous reaction of nitrogen oxides, and rephrased in the revised manuscript.
- Line 51-54: Recently, as $O_3$ has increased along with a decrease in $NO_x$ emission over many regions including East Asia, interest in the heterogeneous reaction of reactive nitrogen oxides, which is yet to be understood, has been newly raised.

l.43: to the uniniated reader it might not be clear what heterogeneous reactions have to do with NOy and ozone chemistry. This would merit one or a few more general sentence(s) to describe NOy chemistry. If this text will get a little longer, please consider sumamrizing the HONO/NOy chemistry in a supplement and refer to it. Nevertheless, one or two sentences will be needed here.

- As suggested, the background information is added to the Introduction in the revised manuscript.
- Line 57-69: In particular, there are growing number of evidence for heterogeneous formation of HONO in relation to high PM2.5 and O3 occurrence in urban areas (e.g., (Li et al., 2021b)). As an OH reservoir, HONO will expedite the photochemical reactions involving VOCs and NOx in the early morning, leading to O3 and fine aerosol formation. Nonetheless, its formation mechanism has not been elucidated clearly enough to be constrained in conventional photochemical models. In addition to the reaction of NO with OH (Bloss et al., 2021), various pathways of HONO formation have been suggested from laboratory experiments, field measurements and model simulations: direct emissions from vehicles (e.g., (Li et al., 2021a)) and soil (e.g.,(Bao et al., 2022)), photolysis of particulate nitrate (e.g., (Gen et al., 2022)), and heterogeneous conversion of NO2 on various aerosol surfaces (e.g., (Jia et al., 2020)), ground surface (e.g.,(Meng et al., 2022)), and microlayers of sea surface (e.g., (Gu et al., 2022)). Among these, heterogeneous reaction mechanism at surface is major concern in recently HONO study.

l.44: you could add https://doi.org/10.5194/acp-18-3147-2018 to the list of references here.

- It is cited as follows.
- Line 54-56: Currently, the lack of measurement of individual $NO_y$ species hindered a comprehensive understanding of the heterogeneous reactions (Anderson et al., 2014; Wang et al., 2017b; Chen et al., 2018b; Stadtler et al., 2018)

l.52/53: it would be useful to know if there is general agreement among these different measurement methods or if they haven't reached a satisfactory level of consistency yet. In the following sentence, please provide some order of magnitude numbers of observed versus simulated HONO levels (or a value range).

- In several intercomparison studies (Pinto et al.,2014; Yi et al., 2021; Yang et al., 2022), all instruments showed reasonable performance with their inherent weaknesses, depending on conditions such as meteorology,

pollution levels, and so on. In general, however, QC-TILDAS was accepted as a reference method with which all measurements from different techniques were compared.

- The calculated HONO from model explains at most 60~90 % of the observed.

- Line 75-79: Of these methods, QC-TILDAS has served as a reference for intercomparison of measurement data from different techniques due to high time resolution and stability (Pinto et al., 2014). In comparison, the model captured at most 67~90 % of the observed HONO in megacities such as Beijing (Tie et al., 2013; Liu et al., 2019)

l.57: the recent adaptation of machine learning techniques in atmospheric sciences is more general that "multi layer artificial neural network". In this context, it suffices to say that "machine learning" has been adopted. Then, in a following sentence you can narrow this down to the employment of deep (artificial) neural networks, which have a capability to learn more complex non-linear relations in data, but also require larger amounts of data for training." The selection of references appears a bit arbitrary. For example, there is a whole special issue in Philosophical Transactions A () on machine learning for weather and climate. Indeed, you may want to first provide two or three general references for ML in atmospheric science (with cf.), then write a sentence which refers specifically to atmospheric chemistry/atmospheric composition and provide some more references there.

- Thank you for detailed advice. This part is rewritten as suggested.
- Line 81-95: In recent years, Machine Learning (ML) method has been adopted in the atmospheric science for pattern classification (e.g. New Particle Formation event) and forecasting and spatiotemporal modelling of $O_3$ and $PM_{2.5}$ (Arcomano et al., 2021;Shahriar et al., 2020;Krishnamurthy et al., 2021;Cui and Wang, 2021;Joutsensaari et al., 2018;Chen et al., 2018a;Kang et al., 2021). Among ML methods, the Neural Network (NN) architecture is widely used owing to its powerful ability to process large amounts in data, allowing improvement in the performance of conventional models through being integrated with physical equations (Reichstein et al., 2019;Schultz et al., 2021). As a NN architecture, a multi-layer artificial neural network, referred to as a Deep Neural Network

(DNN), employs a statistical method that learn non-linear relations of data and obtain the optimum solution for the target species without prior information on the physicochemical processes. DNN has advantages over other NN architecture such as Convolution NN (CNN) or Long-Short Term Memory (LSTM) because it works well for discrete spatiotemporal data. In general, the performance of DNN is similar to or better than other ML methods for small number of data as well as large data set (Baek and Jung, 2021;Dang et al., 2021;Sumathi and Pugalendhi, 2021).

l.59-62: the description why deep learning might be useful for the analysis of atmospheric chemical measurements remains vague and superficial. You should state explicitly that neural networks learn relations in data (similar to function fitting) and you should state in what way NNs may improve on numerical simulations (I guess you refer to the fact that they are inherently bias-free?).

- The NN architecture has advantage in handling the data which has non-linear relation between dataset. Also it shows good performance when the information of physicochemical process is not clear. And finally, the result from NN architecture can be used to numerical models as input data, and it can contribute to the improvement of prediction performance indirectly. This part is rewritten as follows.

- Line 88-95: As a NN architecture, a multi-layer artificial neural network, referred to as a Deep Neural Network (DNN), employs a statistical method that learn non-linear relations in data and obtain the optimum solution for the target species without prior information on the physicochemical processes. DNN has advantages over other NN architecture such as Convolution NN (CNN) or Long-Short Term Memory (LSTM) because it works well for discrete spatiotemporal data. In general, the performance of DNN is similar to or better than other ML methods for    small number of data as well as large data set (Baek and Jung, 2021;Dang et al., 2021;Sumathi and Pugalendhi, 2021).

l.62/63: introduction of the model acronym: difficult to disentangle the sentence - see comment on abstract above.

- The full name of the model is separated with quotation marks, and this part is rewritten as follows.
- Line 105-107: In this study, we aimed to construct a user-friendly 'Reactive Nitrogen species simulation using DNN' (RND) model and estimate HONO mixing ratio using routinely measured atmospheric variables in a highly polluted urban area.

l.67: as this is supposed to be a manuscript for the special issue on "machine learning methods and benchmark datasets", you should add a statement here that the code and training data can be downloaded from ..." (you can of course also refer to the code and data availability section here). Re-usability of your model is a key aspect for this special issue (and for GMD in general).

- As suggested, a statement declaring the reusability of our model, is added to the revised manuscript.
- Line 119-120: The dataset used to train-test-validation can be downloaded from Gil et al., 2021.

l.70: the steps which are described don't guide the development of RND, but describe the typical machine learning workflow.

- Yes, these steps are like a general machine learning model construction workflow which is for users and stated in the text.
- Line 115-118: The development of RNDv1.0 model follows the systematic steps similar to a general machine learning model construction workflow, that including collecting data, preprocessing data, building the DNN, training and validating the model, and testing the performance of the model (Figure 1).

l.77: similar issue - this reads as if every user of RND will first have to perform measurements for her/himself. Please separate the dataset preparation from the model development. The model should be generalizable, i.e. be independent of the specific set of measurements which you describe in the paper.

- Yes, you are right. HONO measurement is not a part of dataset preparation for model run, but for model development. To clarify this point, the section title is modified, and a statement is added in the revised manuscript.
- Line 122: 2.1. Collection of measurement data for model construction
- Line 125-126. It is noteworthy that the HONO measurement data is for model construction and is not required to run the RND model.

l.105: "wind direction should be converted..." - please describe what you did, not what should be done.

- This part is reworded.
- Line 154: Wind direction in degrees were converted to a cosine value for continuity

l.106: "missing values" same as above. Did you filter or interpolate?

- This part is also reworded. They were filtered.
- Line 153-155: For model operation, data of all variables must exist in each hourly data set. So we conducted data integrity test, and filtered the hour array where missing values are exist.

l.107: what is an "array of measurement data"? Also, what is missing is a description of the time resolution of the measurements and how many independent samples were prepared for the machine learning. How was the train-test-val split done? Have you checked the frequency distributions of the (normalized) variables? Have you considered log transform for non Gaussian variables? How many time steps are included in each sample?

- Actually, 'array' is not necessary in this sentence. To avoid confusion, it is removed in the revised manuscript. For input variable, hourly measurements were used.
- The data were split for train, validation, and test, as shown in Figure 3: Data obtained during May-June in 2016 and in 2019 were used for train, June 2018 were used for validation, and April 2019 were for test. The

number of train, validation and test data are 1122, 381, and 133, respectively, which is stated in chapter 2.4.

- The input data were normalized using min-max scaling method, of which frequency distributions are presented below. Other normalizing method can change the distribution of data set, therefore we used min-max scaling method in this study to preserve its original distribution.

- Line 151: As input variables, hourly measurements of chemical and meteorological parameters are used,

- Line 185-187: The RNDv1.0 model was trained, validated, and tested with HONO measurements obtained during May ~ June in 2016 and 2019, in June 2018, and in April 2019, respectively (Figure 3). The number of data used for train, validation, and test were 1122, 381, and 133, respectively.

[Figure]

Section 2.3: there is a lot of information missing from the network description: how many nodes per layer? What is the learning rate? How many epochs were trained? Did the learning rate change during training? Did you try out different numbers of layers and nodes per layer to determine the optimum model? Did you perform a hyperparameter search? Also, what exactly is the input data and what exactly is the target output? Loss function... Those things are standard in the machine learning literature and should be adhered to. I see some of this information appears in the figures and the following section (varying the number of nodes), but this belongs in the model description text.

- The detail information of network are written in chapter 2.3 and chapter 2.4 of paper (e.g. nodes per layer, learning rate, epochs, ... and etc.). We performed the hyperparameter test to decide the number of hidden layer and nodes, from 1 to 10, and from 1 to 100, respectively. For a simple

structure, the hyperparameters that may not strongly affect the model performance were not seriously searched. The node number of each hidden layer remained the same with the fixed learning rate. The information on hyperparameters including activation function, loss function, and optimizer, and data are stated in chapter 2.3 and chapter 2.4, as well.

l.136: if June 2018 has been used in the training already, then this month is not an ondependent test dataset any more.

- The data from June 2018 were not used for training. For training, the data from May ~ June in 2016 and 2019 were used.

l.154: does this mean that you always used the same number of modes in each layer? And you did not try to reduce the number of layers? 1600 samples appears rather small for a network with 5 layers.

- In previous study, the HONO simulation using a 1-hidden layer model with ~300 x 8 data resulted in the correlation coefficient of ~ 0.7 (Gil et al., 2021). In addition, the highest IOA and lowest MAE were observed for 5 hidden layers when performance test was conducted for 1, 5, and 10 hidden layers (Gil et al., 2020). Based on this result, the node number was searched with 5 hidden layers through k-fold cross validation. The node number searching test was conducted using the same number of nodes in each layer due to the limitation of computational resources.

l.160: I don't understand this. First you train the network for 2016 to 2019, then you run it again to obtain HONO results? You already have them from the training.(?)

- In this study, the amount of measurement data was not large enough to conduct the full train, validation, and test processes. Therefore, we adopted k-fold cross validation, which was used in other machine learning study for HONO (e.g., Cui et al., 2021). In addition, the traditional validation and test were conducted using the data obtained in June 2018 and April 2019 April. These processes of train, validation, and test are described in section 2.4.

- Line 186-188: The RNDv1.0 model was trained, validated, and tested with HONO measurements obtained during May ~ June in 2016 and 2019, in June 2018, and in April 2019, respectively (Figure 3). The number of data used for train, validation, and test were 1122, 381, and 133, respectively.

- Line 223-239: Finally, the RND model was validated and tested against the measurement data obtained in June 2018 and April 2019. The calculated HONO mixing ratios are compared with those measured in Figure 7, and their MAE and IOA are listed in Table 3. The two sets of model performance test showed that the model reasonably traced what was observed. As the validation result of RND, the MAE and IOA of the calculated and measured in June 2018 are comparable to those of 2016~2019 result. However, the MAE and IOA of the April 2019 measurements were relatively poor compared to the validation results. Especially, the MAE of the April 2019 is about twice as high as those of validation.

  In these two test periods, HONO levels were lower than those observed on validation days (Figure 5), and the model tended to overestimate high HONO concentrations. The large discrepancy in April 2019 is probably due to seasonality: the difference in meteorological and chemical regime of the atmosphere. For example, the monthly average temperature, relative humidity, and NO2 mixing ratio of Seoul in 2019 were 12.1 ℃, 50.9 %, and 29 ppbv in April 2019 and 22.5 ℃, 60.6 %, and 21 ppbv in June 2019 (https://cleanair.seoul.go.kr; https://weather.go.kr). Note that the RNDv1.0 model was trained with the 9 variables measured in early summer (Table 2). Therefore, the more measurement data spanning a full year for training, the more accurate the model estimates will be.

l.167: I doubt that the inability fo the model to capture minima and maxima is due to the limited amount of data. This is a general aspect of regression models and extensively discussed in Kleinert et al (2021): https://doi.org/10.5194/gmd-14-1-2021

- As shown in frequency distribution of input and output variables, the number of high HONO cases are much less than those of low concentrations. Therefore, the ability of model to capture minima and

maxima will be improved with the large amount of data. We hope that recent observations will be incorporated into the model to improve the results.

l.205 and following: this discussion of atmospheric chemsitry doesn't belong into a section describing the application of the model. Is this supposed to be a general discussion section, comparing RND to other (CTM) models?

- As HONO is not properly simulated in general CTM models, their performance could be improved with HONO provided by the RND model. In this section, an example is presented, highlighting the contribution of the RND model rather than an introduction to its practical application.

l.235 Finally, here is a list of the input variables. But is has not been discussed, which variable has which influence on the results. I have a suspicion that the network really makes use only of 3 or 4 of the 9 variables it is given. See Kleinert et al. (2021) for a way how this can be tested with bootstrapping.

- The bootstrap test similar to Kleinert et al. (2021) was conducted by setting each variables to zero with keeping other values and the results were compared with measurements. Among nine input variables, NO2 was found to have the most significant influence on HONO concentration, followed by RH, temperature, and solar zenith angle (Table S1 2). This result is in good agreement with our previous study (Gil et al., 2021) and added to the text and supplementary.

- Line 240-252: 2.5. Influence of input variables to HONO concentration

    Additionally, a simple bootstrapping test was conducted by setting each variable to zero with keeping other variables (Kleinert et al., 2021). Then, the importance of each input variable to HONO concentration was evaluated using MAE and root mean square error (RMSE). Of nine input variables, NO2 was found to have the most significant influence on HONO concentration, followed by RH, temperature, and solar zenith angle (Table S2). The result of bootstrap test is in good agreement with those from our previous study (Gil et al., 2021), where more detailed information such as aerosol surface area and mixing layer height were incorporated into the model and highlighted the role of precursor gases and heterogeneous

conversion in HONO formation. Therefore, these results demonstrate that the RND model constructed using routinely observed variables, reasonably traced the level of HONO in urban atmosphere.

Table S1 The result of bootstrap test using model validation data. The higher errors imply the higher degree of influence.

| Variables (X) | MAE | RMSE |
|---|---|---|
| - | 0.28 | 0.38 |
| $O_3$ | 0.29 | 0.39 |
| $NO_2$ | 0.59 | 0.85 |
| CO | 0.37 | 0.52 |
| $SO_2$ | 0.34 | 0.46 |
| SZA | 0.41 | 0.60 |
| T | 0.52 | 0.68 |
| RH | 0.52 | 0.72 |
| WS | 0.34 | 0.48 |
| WD | 0.29 | 0.39 |

l.250/251: the ML model doesn't gain any physical understanding of the HONO chemistry, so it cannot be used to test the existing knowledge. You could use such a tool to forecast HONO levels, for example to determine if it might be worthwhile conducting HONO measurements at a specific location or during a specific time period. You may also be able to use the tool in the context of quality controlling the measurements: any strong disagreement would raise a warning that measurements should be checked with extra care. Also, you can of course use it to estimate HONO concentrations when these were not measured in order to then perform 0D model runs, as you show in Figure 8. And in this light, I would agree with the statement that RND is a "supplementary tool".

- Thank you for sharing ideas. The detail application of the RND is added to the revised manuscript.

- Line 310-313: Nevertheless, the HONO concentration produced from RNDv1.0 with routine measurements provides the benefit of relatively inexpensive test for measurement quality control, location selection, and supports the data used for traditional chemistry model based on the current knowledge of the urban photochemical cycle.

l.262: please provide an explicit URL here (you can still add the reference)

- We update the DOI of reference (Gil, J.: RNDv1.0 and example, https://doi.org/10.5281/zenodo.5540180, in, Zenodo, 2021)

Technical corrections:

l.55: related to the comment on l.43: you presume that the reader is familiar with the basics of HONO chemistry, but this cannot be taken for granted.

- More detailed HONO chemistry is added to the introduction (Please seed the response to l.43)

l.30 play instead of plays

- It is corrected (l.46)

l.34 and *it* determines...

- It is corrected (l.50)

Citation: https://doi.org/10.5194/gmd-2021-347-RC1

Reference in answers

Gil, J., Kim, J., Lee, M., Lee, G., Ahn, J., Lee, D. S., Jung, J., Cho, S., Whitehill, A., Szykman, J., and Lee, J.: Characteristics of HONO and its impact on O3 formation in the Seoul Metropolitan Area during the Korea-US Air Quality study, Atmospheric Environment, 2021, https://doi.org/10.1016/j.atmosenv.2020.118182., 2021.

Pinto, J., Dibb, J., Lee, B., Rappenglück, B., Wood, E., Levy, M., Zhang, R. Y., Lefer, B., Ren, X.

R., and Stutz, J.: Intercomparison of field measurements of nitrous acid (HONO) during the SHARP campaign, Journal of Geophysical Research: Atmospheres, 119, 5583-5601, 2014.

Cui, L., and Wang, S.: Mapping the daily nitrous acid (HONO) concentrations across China during 2006-2017 through ensemble machine-learning algorithm, Science of The Total Environment, 147325, 2021.

Kleinert, F., Leufen, L. H., and Schultz, M. G.: IntelliO3-ts v1. 0: a neural network approach to predict near-surface ozone concentrations in Germany, Geoscientific Model Development, 14, 1-25, 2021.

---

## Author Comment (AC2)

0. In this paper, deep neural network based model is used to calculate nitrous acid (HONO) mixing ratios based on the analysis using HONO measurement data from Seoul between 2016 and 2019. Since I am not an expert in atmospheric sciences, but in data and computer science, I will in my review focus on the computational method used and its validity based on the size and type of the data.

- Thank you for your constructive comments and helpful advice. The point-by-point responses are given below, along with relevant parts of the revised manuscript, where all changes are marked in blue.

1. The paper is generally well written and takes action to document the use of the suggested model. The citation to code availability is missing DOI (and one has to go over to Zenodo to locate the code)

- The DOI of reference is updated (Gil, J.: RNDv1.0 and example, https://doi.org/10.5281/zenodo.5540180, in, Zenodo, 2021)

2. The approach taken is motivated by the success of deep learning based methods in various areas. However, here (as often elsewhere) it is not taken into account, that deep learning is most useful in situations in which there are massive amounts of training data — which is not the case here. There are nine input features and there are 1636 data items (1122 for training and 514 for validation). Hence, the data is not really massive and because the amount of interactions is limited (only nine input variables), its is quite likely that more traditional machine learning methods would work well (e.g., ordinary linear regression could be used to provide a baseline (and could even suffice), then one could see how e.g., support vector machine or random forest would work). In the paper, the use of deep neural networks is argued by them being more useful than traditional models, because they are able to handle large amounts of data. For the data used, there is no reason to assume that it could not be handled using also some of the traditional methods, in particular, when the data is small, more complicated models are quite prone to overfitting.

Suggestion for improvement 1: Test different ML learning models to be able to evaluate properly the usability of the suggested model.

- You are absolutely right. In general, the performance of deep learning (DL) is better than or at least similar to traditional machine learning (ML) such as support vector machine or random forest (Sumathi et al., 2020; Baek et al., 2021). This advantage would be greater with larger data set and even small

data set can benefit from it (Dang et al., 2020). DL is also known to be better than general liner regression for data in non-linear relationship.

The test result of RNDv1.0 demonstrates that it reasonably represents ambient HONO levels and captures well the averaged variation. In comparison, it tends to underestimate high concentrations. This is a weakness of our model but indicates that our model does not overfit the training dataset.

In the revised manuscript, introduction is fully revised with background information on HONO and the application of DNN to HONO simulation.

Line 85-104:   Among ML methods, the Neural Network (NN) architecture is widely used owing to its powerful ability to process large amounts of data, allowing improvement in the performance of conventional models through being integrated with physical equations (Reichstein et al., 2019;Schultz et al., 2021). As a NN architecture, a multi-layer artificial neural network, referred to as a Deep Neural Network (DNN), employs a statistical method that learn non-linear relations in data and obtain the optimum solution for the target species without prior information on the physicochemical processes. DNN has advantages over other NN architecture such as Convolution NN (CNN) or Long-Short Term Memory (LSTM) because it works well for discrete spatiotemporal data. In general, the performance of DNN is similar to or better than other ML methods for small number of data as well as large data set (Baek and Jung, 2021;Dang et al., 2021;Sumathi and Pugalendhi, 2021).

When the DNN method is applied to atmospheric chemical constituents, it requires large amount of data for training and thus, the size of measurement data becomes a limiting factor for trace species such as HONO, which are not routinely measured such as O3 or PM2.5. In this regard, the daily average HONO mixing ratio was attempted to be estimated using ensemble ML models with satellite measurements (Cui and Wang, 2021). In comparison, the hourly HONO mixing ratio was calculated using a simple NN architecture with measured variables, which were thought to be closely linked with HONO formation (Gil et al., 2021). The accuracy of the hourly HONO estimated from input variables such as aerosol surface areas and mixed layer height was better than the daily HONO estimate.

3. My second concern is the feature selection or the lack of it. The model blindly uses the nine input variables from the data. This kind of "taking an ML model off-the-shelf" very rarely produces the best possible results and can seriously affect the performance of the model. In addition to feature selection, it might be also possible to compute some surrogate features, e.g., provide information about dependencies in the modelling domain, reducing the need for the ML models to explicitly model these dependencies.

Suggestion for improvement 2: Use feature selection (for all the models) to search for a best possible set of input features.

- The OH produced from HONO photolysis will fuel the photochemical formation of O3 and PM2.5, which are target species of 0-dimensional photochemical models and chemical transport models (CTM). It is demonstrated in section 3 that the presence of HONO has a significant contribution to the performance of photochemical model.

  In this regard, the purpose of this study is to construct a model for estimating the HONO mixing ratio using atmospheric variables that are continuously and routinely measured, but not to improve the performance of model in which the accuracy matters. We hope that our recent observations will be incorporated into the RND model, and the model will be able to provide robust HONO concentrations for operational forecasting models in the future.

  In a previous study, we built a simple Neural Network model that estimated HONO mixing ratio, and we know that selecting the appropriate variables can increase the accuracy of the model (Gil et al., 2021). In this study, we aim to construct a kind of universal and cheap model to estimate HONO concentration in urban areas using atmospheric variables provided through measurement networks. These input variables that were used for model construction did not show any meaningful correlations (Figure S2)

  In addition, bootstrap test similar to what was done in Kleinert et al. (2021), was conducted by setting each variable to zero with keeping other values and the results were compared with measurements. Among nine input variables, NO2 was found to have the most significant influence on

HONO concentration, followed by RH, temperature, and solar zenith angle (Table S2). This result is in good agreement with our previous study (Gil et al., 2021), implying that the input feature used for the model are suitable for estimating HONO concentrations.

In the revised manuscript, the detailed feature selection process is stated in Section1 and Section2.

- Line 105-107: In this study, we aimed to construct a user-friendly 'Reactive Nitrogen species simulation using DNN' (RND) model and estimate HONO mixing ratio using routinely measured atmospheric variables in a highly polluted urban area.
- Line 151-154: As input variables, hourly measurements of chemical and meteorological parameters are used, including the mixing ratios of O3, NO2, CO, and SO2, along with temperature (T), relative humidity (RH), wind speed (WS), wind direction (WD), and solar zenith angle (SZA) to estimate the target species, HONO, as the output.
- Line 241-253: 2.5. Influence of input variables to HONO concentration

  Additionally, a simple bootstrapping test was conducted by setting each variable to zero with keeping other variables (Kleinert et al., 2021). Then, the importance of each input variable to HONO concentration was evaluated using MAE and root mean square error (RMSE). Of nine input variables, NO2 was found to have the most significant influence on HONO concentration, followed by RH, temperature, and solar zenith angle (Table S2). The result of bootstrap test is in good agreement with those of our previous study (Gil et al., 2021), where more detailed information such as aerosol surface area and mixing layer height were incorporated into the model and highlighted the role of precursor gases and heterogeneous conversion in HONO formation. Therefore, these results demonstrate that the RND model constructed using routinely observed variables, reasonably traced the level of HONO in urban atmosphere.

4. Finally, the testing of the model using data from April 2019, shows some of the limitations of the developed model. It seems that there is an occurrence of concept drift (when the distribution of data changes, the model does not work well anymore). Also, the error might increase due to overfitting of the model. This aspect should be studied further, in particular it would be important to

be able to provide the region in which the model's accuracy is on an acceptable level. There is a rich body of literature in detecting concept drift (for a survey, e.g., see Zliobaite I., Pechenizkiy M., Gama J. (2016) An Overview of Concept Drift Applications. In: Japkowicz N., Stefanowski J. (eds) Big Data Analysis: New Algorithms for a New Society. Studies in Big Data, vol 16. Springer, Cham. https://doi.org/10.1007/978-3-319-26989-4_4).

Suggestion for improvement 3: Analyse the region in which the proposed model can be expected to work, at least provide some discussion on the effect of overfitting and concept drift and how theses affect the usability of the model.

- Atmospheric parameters including meteorological factors and chemical constituents show clear diurnal variations, especially in urban areas with high anthropogenic emissions. For example, NO2 reached the maximum during the morning rush hour, decreased down to the minimum in the afternoon, and increased at nighttime. This type of variation remained nearly constant through the year with changes in seasonal amplitude depending on emissions and meteorological factors determining the dilution and transport of air pollutants. The variation in O3 is just opposite to NO2.

- Our model was constructed for urban applications. When the model was tested against data obtained April, model uncertainty was increased. Although our model was trained and validated with data obtained during May-June, the variations in input variables for test period were similar to those of train-validation periods. Considering the result of previous study about HONO formation mechanism, the increased model uncertainty could be due to some factors that were not constrained in the model such as aerosol surface areas.

  Therefore, it is quite likely that the increased model uncertainty is not associated with the occurrence of concept drift.

Based on these observations, I would reject the paper in its current form, with the encouragement to resubmit, taking the suggestions for improvement into account.

Citation: https://doi.org/10.5194/gmd-2021-347-RC2

Reference in answers

Baek, W.-K., and Jung, H.-S.: Performance Comparison of Oil Spill and Ship Classification from X-Band Dual-and Single-Polarized SAR Image Using Support Vector Machine, Random Forest, and Deep Neural Network, Remote Sensing, 13, 3203, 20

Sumathi, S., and Pugalendhi, G. K.: Cognition based spam mail text analysis using combined approach of deep neural network classifier and random forest, Journal of Ambient Intelligence and Humanized Computing, 12, 5721-5731, 2021.

Dang, C., Liu, Y., Yue, H., Qian, J., and Zhu, R.: Autumn crop yield prediction using data-driven approaches:-support vector machines, random forest, and deep neural network methods, Canadian Journal of Remote Sensing, 47, 162-181, 2021.21.

Cui, L., and Wang, S.: Mapping the daily nitrous acid (HONO) concentrations across China during 2006-2017 through ensemble machine-learning algorithm, Science of The Total Environment, 147325, 2021.

Gil, J., Kim, J., Lee, M., Lee, G., Ahn, J., Lee, D. S., Jung, J., Cho, S., Whitehill, A., Szykman, J., and Lee, J.: Characteristics of HONO and its impact on O3 formation in the Seoul Metropolitan Area during the Korea-US Air Quality study, Atmospheric Environment, 2021, https://doi.org/10.1016/j.atmosenv.2020.118182., 2021.

Kleinert, F., Leufen, L. H., and Schultz, M. G.: IntelliO3-ts v1. 0: a neural network approach to predict near-surface ozone concentrations in Germany, Geoscientific Model Development, 14, 1-25, 2021.

---

## Author Response (AR2)

Correspondence to editor's comments

Thank you for constructive comments, for which responses are given with the relevant part of the revised manuscript. In the revised manuscript, changes are marked in blue.

The revised version shows several improvements, in particular when it comes to addition of clarity, both in terms of writing as well as presenting details of the developed model.

However, one of the main criticisms pointed out by myself, and the other reviewer (in their words "There is no proof of the generalisation capability of the model, so it may well be that this model fails completely if it were applied to measurement data obtained under different conditions.") is still not addressed at all. That is to say, the authors have developed a model (which is fine), but there is no way of knowing how well it generally performes. Since the idea is to develop a model for others to use (of the shelf), it should be made very precise what are the capabilities and restrictions of using the developed model.

Furthermore, I remain sceptical that deep learning provides here significant gains in model accuracy. Obviously, I might be wrong but based on the paper there is no way of telling how good the developed model is. The training data is small, and hence it would be very easy to train some other standard ML models and compare the performance of RND1.0 to those. The authors state that deep learning has previously shown to perform similarly or better than other ML methods, but I see no reason not to compare the developed model to some baseline models and gain some further perspective on the performance of the model.

1. We understand your concern. According to your comments, the uncertainty of RNDv1.0 associated with input variables is estimated from bootstrapping test (Table 4) and discussions are added to the revised manuscript.

In the revised manuscript, the relevant parts are as follows.

**Line 235-248:**

2.5. Influence of input variables on HONO concentration

A simple bootstrapping test was conducted to evaluate the relative importance of the input variable to HONO concentration. In this analysis, each variable was set to zero and MAE was calculated as an evaluation metrics (Kleinert et al., 2021). Of nine input variables, NO2 was found to have the most significant influence on HONO concentration, followed by RH, temperature, and solar zenith angle (Table 4). The highest MAE of 0.59 ppbv can be considered as the maximum uncertainty of RNDv1.0 due to the input variable.

The result of bootstrap test is in good agreement with those of our previous study (Gil et al., 2021), where more variables such as aerosol surface area and mixing layer height were incorporated into the model, highlighting the crucial role of precursor gases and heterogeneous conversion in HONO formation. Therefore, these results demonstrate that the RND model constructed from routinely measured criteria pollutants and meteorological parameters sufficiently captured the HONO variability in the urban atmosphere.

**Line 261-269:**

It is possibly due to the variability of HONO that is not fully captured by RNDv1.0 using 9 input variables. As stated above, heterogeneous reactions intimately involved in HONO formation are not considered in RNDv1.0. More importantly, the annual variability of criteria pollutants such as $PM_{2.5}$ has increased in recent years. Particularly in 2019, the monthly average $PM_{2.5}$ mass concentration was lower in April (21 μg $m^{-3}$) than in May (29 μg $m^{-3}$), unlike normal years. Given that the test result is within the uncertainty range of the model that is primarily determined by $NO_2$ (Table 4), RNDv1.0 will be applicable to urban environments under various conditions.

**Table 4.** The result of bootstrap test of measurement data used to train RNDv1.0 model. The greater the MAE, the greater the influence of variable.

| Variable (X) | MAE (ppbv) |
|:---:|:---:|
| - | 0.28 |
| $O_3$ | 0.29 |
| $NO_2$ | 0.59 |
| CO | 0.37 |
| $SO_2$ | 0.34 |
| Solar zenith Angle (SZA) | 0.41 |
| Temperature (T) | 0.52 |
| Relative humidity (RH) | 0.52 |
| Wind speed (WS) | 0.34 |
| Wind direction (WD) | 0.29 |

2. As pointed out, the performance check of RNDv1.0 is necessary, especially due to the small data set for training. Given that HONO observations are extremely limited, however, we had no choice but to seek for practical evidence. Fortunately, HONO simulations in CMAQ model are available for the KORUS-AQ campaign during May~June 2016, and the two sets of HONO concentrations are compared in the revised manuscript (Figure 7). Figure 7 clearly shows that the conventional chemical transport model, CMAQ, fails to capture variations in the ambient HONO mixing ratio, whereas RNDv1.0 works reasonably (IOA = 0.90 and MAE = 0.3 ppbv for RNDv1.0, and IOA = 0.44 and MAE = 0.8 ppbv for CMAQv5.3.1).

In our previous study, HONO was estimated using a 1-layer ANN model with more input variables such as NO mixing ratio, boundary layer height, and surface area of sub-micron

particles (Gil et al., 2021). The results of RNDv1.0 was then compared to those of ANN model (Figure A below). While the simulations of ANN model tended to trace better the observed HONO concentrations than RND model, the performance of RNDv1.0 was better than that of ANN model ($r^2$ = 0.7 and MAE = 0.3 ppbv for RNDv1.0, and $r^2$ = 0.6 and MAE = 0.4 ppbv for ANN model) (Gil et al., 2021). The accuracy of RNDv1.0 is also better than that of ensemble ML models (RF, BPNN, GBDT, $r^2$ = 0.7, MAE = 0.3 ~ 0.5 ppbv) (Cui et al., 2021).

In the revised manuscript, the relevant parts are as follows.

**Line 226-233:**

Next, the HONO calculated in RNDv.1.0 was compared with observations and results from CMAQ (Community Multi-scale Air Quality Model, v5.3.1) simulations during the KORUS-AQ study (May~June 2016) (Figure 7). More information on CMAQ modeling can be found elsewhere (Appel et al., 2021). While the results of RNDv.1.0 reasonably traced the observed variations (IOA = 0.90), the CMAQ severely underestimated the measured HONO concentration (IOA = 0.44). These results demonstrate the performance and efficacy of RNDv1.0 in calculating the ambient HONO mixing ratio that are poorly reproduced in conventional operating models.

[Figure]

Figure 7. During the KORUS-AQ campaign (May-June 2016), HONO mixing ratios calculated using RNDv1.0 (red dot) are compared with those observed (gray square) and calculated using CMAQv5.3.1 (blue cross).

[Figure]

Figure A. HONO concentrations calculated in RNDv1.0 (red dot) are compared with observations and those calculated in one-layer ANN model (orange cross) and in CMAQv5.3.1 (blue triangle) during the KORUS-AQ campaign (May~June 2016).

---

## Author Response (AR3)

**Correspondence to editor's comments**

I have gone through your response and it does yet answer the concerns of the reviewers. Both reviewers have raised the issue with generalization: how well would the model perform in other conditions. Your validation and testing datasets are very small covering quite narrow range of meteorological conditions. I understand that the amount of data you have is limited but still the concerns raised by the reviewers should be answered before the manuscript can be published in GMD. Now from the manuscript it appears that you have developed deep-learning model to your data but it does not show that you would have developed a model which is applicable elsewhere.

In 2021, we obtained additional measurement data sets during May-Jun and Oct-Nov, which were used to test the RNDv1.0. Therefore, the RND model was tested on measurements acquired in weather conditions different from those of the train dataset (Figure 3). The test results are presented in Figure 7(a): IOA = 0.68, MAE = 0.74, r = 0.55, and RMSE = 0.95. When the data in which at least one input parameters do not fall within the range of the train dataset is excluded from the test dataset, there is no significant difference in the performance of RNDv1.0 between the two that meet same atmospheric conditions or do not meet the criteria (Figure S5 and Table S2).

It is particularly noteworthy that severe haze pollution events occurred in November 2021, when the daily average $PM_{2.5}$ concentration was raised up to 120 $\mu g\ m^{-3}$ and the HONO mixing ratio also increased to 4 ppbv or more in Seoul. Except for these extremes, RNDv1.0 traces well the variation of HONO mixing ratio.

It is good that you have tested the 1-layer ANN model. But in order to answer the reviewer concern you need to test also some simpler ML model(s) and add those to the manuscript. In you response you show comparison of you model and ANN, but this is for the training data. In general showing good correspondence with the training data (Figures 5-7 in the manuscript) does not tell how the model performs for "independent" datasets. Thus, after you have conducted additional simulations you should improve the model performance analysis with non-training data with proper scatter plots (similar to Fig A in the referee response) and statistical information (not just MAE and IOP but also RMSE, R). In addition, you need to answer the reviewer comment "Since the idea is to develop a model for others to use (of the shelf), it should be made very precise what are the capabilities and restrictions of using the developed model" by clearly stating the limitations and benefits of the model.

Pre-constructed 1-layer ANN model needs additional input parameters (boundary layer height and aerosol surface area), and unfortunately these data are not exist on test periods. Therefore as recommended, a random forest (RF) model was constructed using the same data and process of the RNDv1.0 construction and its results were compared with those of RNDv1.0, CMAQv5.3.1, and 1-layer ANN for the measurement data from 2016 KORUS-AQ campaign (Figure 5 & 6 and Table 3), and also for the test data (Figure 7). We are agreeing about your concern that "train" data should not be used to evaluate model performance in generally, so the comparison using 2016 measurement data was become a part of train-validation process (Figure 3).

Recently, we acquired HONO measurement data during May~Jun and Oct~Nov in 2021, so these data set are added in the test data set (Figure 3). By using this test data set which 2021 observation data added, the performance of RND1v1.0 and RF model were evaluated (Figure 7). The performance evaluation results using the test dataset, and the bootstrap (BS) test results of RNDv1.0 and RF clearly demonstrated that the ability of the deep learning model to simulate the HONO mixing ratio is more adequately in the urban atmosphere compared to the general machine learning model (Table 4). Statistical information including RMSE and r is provided for model evaluation (Figure 7 and Table 3).

In addition to these, the manuscript should be checked by language services as there are several issues with the language. In addition, at the end of page 7, the sentence on line 204 is unfinished.

The manuscript was thoroughly checked, and errors were corrected.

[Figure]

**Figure 3.** Design of training, validation, and test to build RNDv1.0 using measurement data. The k-fold cross validation was performed using randomly divided five subsets of training data set.

[Figure]

**Figure 5**. Comparison between measured HONO (HONO$_{obs}$) and calculated HONO (HONO$_{mod}$) using CMAQv5.3.1 (blue triangle), RF (purple square), ANN (orange star), and RNDv1.0 (red circle) during the KORUS-AQ campaign (may-June 2016)

[Figure]

**Figure 6**. Average diurnal variation of measured HONO (HONO$_{obs}$) and calculated HONO (HONO$_{mod}$) using CMAQv5.3.1 (blue triangle), RF (purple square), ANN (orange star), and RNDv1.0 (red circle) during the KORUS-AQ campaign (may-June 2016)

[Figure]

**Figure 7**. Relationship between measured HONO (HONO$_{obs}$) and modeled HONO (HONO$_{mod}$) using (a) RNDv1.0 and (b) a Random Forest model for the test dataset.

[Figure]

Figure S5. Relationship between measured HONO (HONOobs) and modeled HONO (HONOmod) using RNDv1.0 (red) and a Random Forest (purple) for the test dataset. (a) and (b) present data in which all input variables are within the range of the train dataset, and (c) and (d) are the others that do not meet the criteria.

**Table 1.** The performance of chemical transport model (CMAQv5.3.1) and machine learning (ML) models including Random Forest (RF), Artificial Neural Network (ANN), and RNDv1.0 on measurement data from 2016 KORUS-AQ campaign that were used for training.

|      | CMAQv5.3.1 | RF   | ANN  | RNDv1.0 |
|------|------------|------|------|---------|
| IOA  | 0.44       | 0.99 | 0.86 | 0.9     |
| r    | -0.07      | 0.99 | 0.81 | 0.84    |
| MAE  | 0.82       | 0.1  | 0.38 | 0.27    |
| RMSE | 1.06       | 0.12 | 0.41 | 0.37    |

**Table 4.** The result of bootstrap test of measurement data used to train the RF and RNDv1.0 model. The greater the MAE, the greater the influence of variable.

| Variable | RF | | RNDv1.0 | |
|---|---|---|---|---|
| | MAE | Feature Importance | MAE | Feature Importance |
| - | 0.10 | - | 0.28 | - |
| $O_3$ | 0.57 | 1 | 0.29 | 8 |
| $NO_2$ | 0.24 | 4 | 0.59 | 1 |
| CO | 0.19 | 7 | 0.37 | 5 |
| $SO_2$ | 0.17 | 8 | 0.34 | 6 |
| Solar zenith Angle (SZA) | 0.25 | 2 | 0.41 | 4 |
| Temperature (T) | 0.21 | 5 | 0.52 | 2 |
| Relative humidity (RH) | 0.25 | 3 | 0.52 | 2 |
| Wind speed (WS) | 0.20 | 6 | 0.34 | 6 |
| Wind direction (WD) | 0.13 | 9 | 0.29 | 8 |

**Table S2**. The performance of RNDv1.0 and a Random Forest (RF) model on the test dataset that is divided into 'in' where all input parameters fall within the range of the train dataset and 'out' that do not meet the criteria.

|      | RNDv1.0_in | RF_in | RNDv1.0_out | RF_out |
|------|-----------|-------|-------------|--------|
| IOA  | 0.71      | 0.28  | 0.82        | 0.73   |
| r    | 0.55      | -0.02 | 0.52        | 0.10   |
| MAE  | 0.64      | 0.55  | 0.63        | 0.66   |
| RMSE | 0.86      | 0.87  | 0.96        | 1.24   |

**The revised parts are as follows.**

**Line 24:**

~ the several months from 2016 to 2021.

**Line 25-27:**

[revised manuscript text omitted]

---

## Author Response (AR4)

**Response to Reviewer 2**

Thank you very much for your constructive and critical comments. Accordingly, this manuscript has been revised. Regarding the response, please note that the line numbers in your review do not exactly match with those in the latest manuscript and thus, responses are given based on comments.

Gil et al present a model based on Deep Neural Networks for estimation of HONO concentrations in urban environments using measurements of classical atmospheric pollutants and meteorological variables as input. Because HONO measurements are hardly available, the authors argue that using estimated HONO as input in photochemical models improves the calculation of the OH production rate and of O3 concentrations. This is an interesting and valuable piece of work, however, there are a few issues that should be resolved before this manuscript can be published in Geoscientific Model Development.

My main concern is the way the performance of the RND v0.1 model is evaluated and the conclusions and recommendations that are drawn from the model performance evaluation. For performance testing, both, the training set and the testing set are used. This is not correct, the training and validation data should only be used for model building, the performance assessment should only be done based on the test data. It is found that the model performance is much better for the period used for training and validation than for the test data. This is of course not surprising and indicates clear limitations of the model (e.g. over-fitting). The model performance assessment needs to be changed accordingly.

You are absolutely right that the data set used to train the model should not be re-used in the testing process. As illustrated in Figure 3, we divided the observation data into a training set (May-June 2016, May-June 2019) and a testing set (June 2018, April 2019, May-June 2021, and Oct-Nov 2021). This fact is more clearly expressed in the revised manuscript as follows.

> Line 220-222: RNDv1.0 and the RF model were tested using data obtained in June 2018, April 2019, and May–June 2021 and October–November in 2021, which were not used for RNDv1.0 training (Figure 3).

[Figure]

**Figure 3.** Training, validation, and test design to build RNDv1.0 using the measurement data. The k-fold cross validation was performed using randomly divided five subsets of the training data set.

The model performance was particularly poor for the test data from April 2029. The authors explain this by the fact that the conditions during April 2019 were different from the conditions covered by the training data. This points to another important aspect that is entirely neglected in the current manuscript: What are the conditions the RND v0.1 model can be applied with a performance as determined? What happens when the model is applied to conditions that are not covered by the training data (model applied to meteorological conditions and/or atmospheric pollutant concentrations outside the range covered in the training data)? It is very likely that applications of the proposed DNN model at other locations and during other times of the year will face this situation. It is necessary that this issue is addressed.

The authors say in the abstract and in the introduction section that the RND v0.1 model is proposed for calculation of HONO mixing ratios in highly polluted urban environments. In the results section, the model is described as being fit for application in any urban area (page 6, line 172). The conditions (in terms of air pollutant concentrations) where RND v0.1 can be applied should be made more clear.

This manuscript has been revised through several rounds of review, during which the recent HONO measurements performed in May-June and Oct-Nov 2021 were added to the test data set. In the new test set, extreme events that occurred in early winter are included. As a result, significant revisions were made to the manuscript, including sections "2.5. Model test" and "2.6. Bootstrap test and feature importance".

In this study, testing data was obtained in a wider range of ambient air conditions than training data. It provides a good opportunity to evaluate the applicability of RNDv1.0. First, the performance of RNDv1.0 was compared with those of other models in Section 2.5. Then, the influence of input variables on the output results was analyzed in Section 2.6. In addition, detailed information is provided in Supplementary Information.

Actually, the extreme conditions are hardly constrained by model, causing the discrepancy between the measurements and model results. It is especially the inherent limitation of data-driven model. Given this fact, the RNDv1.0 is able to trace the variation of HONO when applied in the range of training data set, which is still difficult in other conventional models. We therefore suggested the conditions for the application of RNDv1.0 clearly as our training data coverage, and warning about its use in the different conditions. Consequently, it is reasonable to argue that this study demonstrates the applicability of RNDv1.0 to urban atmospheres enriched with $NO_x$.

The relevant parts are revised as follows.

> Line 228-244: The performance of RNDv1.0 was slightly lower than that of the RF model, but it well traced the HONO mixing ratio. Among the test dataset, the early winter (October–November) data are particularly valuable for demonstrating the applicability of RNDv1.0 because they stem from different weather conditions than the training dataset. For example, HONO mixing ratios reached over 4 ppbv when the daily average $PM_{2.5}$ concentration increased to 120 μg m$^{-3}$ during severe haze pollution events. Therefore, in the next step, the performance of RNDv1.0 was compared for the two cases by dividing the test dataset into a group in which all input variables fall within the range of the train dataset and a group which does not meet this criterion. In RNDv1.0, there was no significant difference in performance between the two groups (Figure S5 and Table S2). When the data in which at least one input variable does not fall within the range of the training dataset were excluded from the test dataset, no significant difference was observed in the performance of RNDv1.0 between the two that meet same atmospheric conditions or do not meet the criteria (Figure S5 and Table S2). These extreme atmospheric conditions can make the model performance be worsened. Except for these extremes, RNDv1.0 well traced the

variation of the HONO mixing ratio. These results demonstrate the applicability of RNDv1.0, which is not strictly constrained by atmospheric conditions. The influence of input variable are further analyzed in the next section.

Line 314-324: RNDv1.0 was constructed using the measurements made in a high $NO_x$ environment where the maximum $NO_2$ reached about 80 ppbv. During the measurement period, the HONO mixing ratio was increased up to about 7 ppb under the influence of air masses originating from China. When applying RNDv1.0 to regions or times heavily affected by transport, the model could possibly underestimate the HONO level without more detailed information, such as nanoparticles. Indeed, a previous study showed that HONO formation is closely related to the surface areas of submicron particles (Gil et al., 2021). Nevertheless, RNDv1.0 is advantageously a relatively inexpensive test for measurement quality control and location selection, and it supports the data used for traditional chemistry models based on the current knowledge of the urban photochemical cycle. Therefore, RNDv1.0 can serve as a supplementary tool for conventional forecasting models.

The paper is generally well written, however, there are rather many small linguistic errors such as missing articles (e.g. page 3, line 84; pg. 5, lines 139 and 140; page 6 line 159) and wrong grammar (e.g. should consequently be "training and validation" instead of "train and validation", and also often "testing" instead of "test". The manuscript should again be carefully checked and corrected.

This manuscript was carefully checked, and errors were corrected.

Other comments:

Page 2, line 54-56: The authors write about "the" model and "this underestimation". It is unclear what model is meant, it seems that it is referred to photochemical models in general. Please make this clear and revise accordingly.

This sentence is rewritten as follows.

Line 74-76: In comparison, the WRF-Chem and RACM2 models captured approximately 67 %–90 % of the observed HONO in megacities such as Beijing

Page 3, line 70, should be "including data collection" instead of "including collecting data".

It is changed as you suggested.

Line 110-112: The RNDv1.0 development follows systematic steps that are similar

to a general ML model construction workflow, including data collection, preprocessing data, building the DNN, training, and validating the model, and testing the model performance.

Page 4, line 95-97. The 10th and 90th percentile mixing ratios for the input variables are given. It is not mentioned what the time basis of these values are, are these hourly or daily values? The temporal resolution should be provided.

Statistics are provided for hourly measurements, which is clearly stated in the revised manscuript.

Line 135-136: The measurement statistics for the entire experimental periods are presented in Table 2 and Table S1.

Table 2. Input variables and their concentrations ($10^{th}$–$90^{th}$ percentile of the hourly measurements), coverage, and scale factors for the RNDv1.0 model. Measurements were conducted in Seoul during May–June in 2016 and 2019.

Table S1. The range of input variables (hourly measurements) used in this study.

Page 4, line 102. Terminology "chemical and meteorological parameters" is not correct here. In the usual convention, the input variables are denoted as "variables" and not as "parameters". The parameters are their weights in a statistical model. Please change.

It is changed through the manuscript.

Line 101-103: This study aims to develop a user-friendly "reactive nitrogen species simulation using DNN" model (RNDv1.0) that estimates the HONO mixing ratios from the real-time measurements of criteria pollutants and meteorological variables.

Line 118-119: To construct RNDv1.0, measurement data were obtained, including HONO, reactive gases, and meteorological variables

Line 131-134: In addition to HONO, trace gases including $O_3$, $NO_2$, CO, and $SO_2$ as well as meteorological variables including temperature (T), relative humidity (RH), wind speed (WS), and wind direction (WD) were measured.

Line 141-144: As input variables, hourly measurements of chemical and meteorological variables were used, including the mixing ratios of $O_3$, $NO_2$, CO, and $SO_2$, along with T, RH, WS, WD, and solar zenith angle (SZA) to estimate the target species, HONO, as the output.

Line 194-196: The high IOA value signifies that the performance of RNDv1.0 is adequate, and it is capable of simulating the ambient HONO mixing ratio using the

routinely measured criteria pollutants and meteorological variables.

Line 199-200: The RF model was constructed using the KFCV method and the same input variables as RNDv1.0

Line 233-237: Therefore, in the next step, the performance of RNDv1.0 was compared for the two cases by dividing the testing dataset into a group in which all input variables fall within the range of the training dataset and a group which does not meet this criterion. In RNDv1.0, there was no significant difference in performance between the two groups

Line 263-265: Thus, it is reasonable to state that RNDv1.0 constructed using routinely measured criteria pollutants and meteorological variables can sufficiently capture the HONO variability in the urban atmosphere.

Page 4, lines 105-107. The authors write that wind direction "should" be converted and there "should" be no missing values. From the text it seems clear that the authors have converted the measured wind direction and they have removed observations with missing values. I think the authors should rephrase the text so that it is clear what data conversion and selection steps have been done.

The relevant part is rephrased as follows.

Line 144-146: The WD in degrees was converted to a cosine value for continuity. In the last step of data processing, hourly measurement sets were removed from the input data set if any of the nine variables were missing.

Page 4, equation 1. I stumbled over the notation F1 and F2. It seems that these are simply the observed min and max of variable x. Why not denoting F1 and F2 as x_min and x_max? Would probably be more clear.

Yes, you are right. For clarity, the relevant part is reworded as follows.

Line 155-157: where $x_{raw}$ is the raw data, $x_{sca}$ is the scaled value, and the scale factors of $F_1$ and $F_2$ correspond to the maximum-minimum and minimum values of the input variable (X), respectively, which are listed in Table 2

---

## Author Response (AR5)

This revised manuscript has clearly improved. The points raised by the reviewers have adequately been addressed. I think that the paper can now be published. I have a few suggestions for technical corrections (see below).

Thank you for your comments. The point-by-point responses are given below.

Abstract, line 15: The "one of" can be deleted. Or write something like "HONO, an oxidized nitrogen compound, which plays an important role ..."

**Revised:** Nitrous acid (HONO) plays an important role in the formation of ozone and fine aerosols in the urban atmosphere

Page 3, lines 82-83. The sentence that is new in the revised version is difficult to understand and probably not correct. Please rewrite.

**Revised:** Among the ML methods, the neural network (NN) architecture is widely used owing to its powerful ability to process large volumes of data. In particular, a multilayer artificial NN (ANN), referred to as a deep NN (DNN), employes statistical methods to learn nonlinear relationships within the data and yield optimal solutions for a target species without prior knowledge of the underlying physicochemical processes (Reichstein et al., 2019;Schultz et al., 2021).

Page 4, line 114. Should be Gil (2021) instead of Gil et al. (2021).

**Revised:** The dataset used to train, test, and validate can be downloaded from Gil (2021).

Page 5, line147. Should be "(2847 observations)" or "(2847 data points)" or similar instead of "(2847)".

**Revised:** Finally, 54.2 % of all the available measurement data (2847 data points) were used to construct and evaluate RNDv1.0.

Page 11, lines 318-320. The role of particles for HONO formation should be made a bit clearer. Avoid to use the two terms "nanoparticles" and "submicron particles" here. They mean not the same and using these different terms here is confusing.

**Revised:** Indeed, a previous study showed that HONO formation is closely related to the surface area of particles with diameters in the range of hundreds of nanometers (Gil et al., 2021).

Figure 6, x-axis. Change label to "Hour" instead of "HONOobs (ppbv)".

[Figure]

**Figure 6**. Average diurnal variation of the measured HONO (HONO_obs) and calculated HONO (HONO_mod) using CMAQv5.3.1 (blue triangle), RF (purple square), ANN (orange star), and RNDv1.0 (red circle) during the KORUS-AQ campaign (May–June 2016).

Figure S4, legend. Should be "trees" instead of "tress".

[Figure]

**Figure S4.** K-fold cross validation for Random Forest (RF) model by changing the number of trees in ensemble.

More general note: The manuscript should again be carefully checked for linguistic errors.